# Representation Balancing with Decomposed Patterns for Treatment Effect Estimation

## Abstract

Estimating treatment effects from observational data is subject to a covariate shift problem incurred by selection bias. Recent research has sought to mitigate this problem by balancing the distribution of representations between the treated and controlled groups. The rationale behind this is that counterfactual estimation relies on (1) preserving the predictive power of factual outcomes and (2) learning balanced representations. However, there is a trade-off between achieving these two objectives. In this paper, we propose a novel model, DIGNet, which is designed to capture the patterns that contribute to outcome prediction (task 1) and representation balancing (task 2) respectively. Specifically, we derive a theoretical upper bound that links the concept of propensity confusion to representation balancing, and further transform the balancing Patterns into Decompositions of Individual propensity confusion and Group distance minimization (PDIG) to capture more effective balancing patterns. Moreover, we suggest decomposing proxy features into Patterns of Pre-balancing and Balancing Representations (PPBR) to preserve patterns that are beneficial for outcome modeling. Extensive experiments confirm that PDIG and PPBR follow different pathways to achieve the same goal of improving treatment effect estimation. We hope our findings can be heuristics for investigating factors influencing the generalization of representation balancing models in counterfactual estimation.

## 1 Introduction

In the context of the ubiquity of personalized decision-making, causal inference has sparked a surge of research exploring causal machine learning in many disciplines, including economics and statistics (Wager & Athey, 2018; Athey & Wager, 2019; Farrell, 2015; Chernozhukov et al., 2018; Huang et al., 2021), healthcare (Qian et al., 2021; Bica et al., 2021a;b), and commercial applications (Guo et al., 2020b;c; Chu et al., 2021). The core of causal inference is to estimate *treatment effects*, which can be tied to a fundamental hypothetical question: What would be the outcome if one received an alternative treatment? Answering this question requires the knowledge of *counterfactual outcomes*, which can only be inferred from observational data, but cannot be obtained directly.

*Selection bias* presents a major challenge for estimating counterfactual outcomes (Guo et al., 2020a; Zhang et al., 2020; Yao et al., 2021). This problem is due to the non-random treatment assignment, that is, treatment (e.g., vaccination) is usually determined by covariates (e.g., age) that also affect the outcome (e.g., infection rate) (Huang et al., 2022b). The probability of a person receiving treatment is well known as the *propensity score*, and the difference between each person's propensity score can inherently lead to a covariate shift problem, i.e., the distribution of covariates in the treated units is substantially different from that in the controlled ones. The covariate shift issue makes it more difficult to infer counterfactual outcomes from observational data (Yao et al., 2018; Hassanpour & Greiner, 2019a).

Recently, a line of representation balancing works has sought to alleviate the covariate shift problem by balancing the distribution between the treated group and the controlled group in the representation space (Shalit et al., 2017; Johansson et al., 2022). The rational insight behind these works is that the counterfactual estimation should rest on **(1)** the accuracy of factual outcome estimation and **(2)** enforcing minimization

of distributional discrepancy measured by the Integral Probability Metric (IPM) between the treated and controlled groups. Wasserstein distance (Cuturi & Doucet, 2014) is the most widely-adopted IPM for the target (2) (Shalit et al., 2017; Huang et al., 2022a; Zhou et al., 2022), whereas other distance metrics such as $\mathcal{H}$-divergence have still received little attention in causal representation learning literature though $\mathcal{H}$-divergence is an important distance metric in other fields (Ben-David et al., 2006; 2010). Unlike previous studies focusing on group distance minimization, we emphasize that the propensity score is a natural indicator of whether representations are adequately balanced. Specifically, when it becomes challenging to distinguish whether each unit in the representation space is treated or controlled, i.e., *propensity confusion*, the representations are believed to be adequately balanced. Ideally, if propensity confusion is extremely strong, i.e., representations are perfectly balanced, the propensity scores of each unit in the representation space would be 0.5. Therefore, propensity confusion provides a logical interpretation for representation balancing, and achieving propensity confusion is naturally connected to minimizing the $\mathcal{H}$-divergence-based individual treatment effect (ITE) error bound. More discusses about the theoretical results and empirical implementations are presented in Section 3.2 and Section 4.1, respectively.

Moreover, a critical issue that remain to be resolved is that enforcing models to learn only balancing patterns can weaken the predictive power of the outcome function. This is because of the trade-off between targets (1) and (2) (Zhang et al., 2020; Assaad et al., 2021; Huang et al., 2022a). We give a motivating example below to help readers better understand this phenomenon. In addition, we provide another illustrative example and analytical explanations in Section A.4 as supplementary elaboration on the aforementioned trade-off.

**Motivating Example.** Consider two individuals who are identical in every aspect except for their age. One person is older and is designated as the treatment (T) group, while the other person is younger and serves as the control (C) group. Age is used as a covariate to distinguish between T and C. If it is known that the older person is more susceptible to a certain disease, the information age (covariate) can be used to predict the likelihood of one developing the disease (outcome). However, suppose the information age of each individual is mapped to some representations such that the representations of T and C are highly-balanced or even identical. In that case, it may be difficult to differentiate between T and C based on these representations. Consequently, these over-balanced representations may lose information to accurately predict the likelihood of each individual developing the disease.

The aforementioned issue motivates us to explore approaches to **(i)** learning more effective balancing patterns (task 2) without affecting factual outcome prediction (task 1) or **(ii)** improving factual outcome prediction (task 1) without affecting learning balancing patterns (task 2). In this paper, we propose a new method, DIGNet, which learns decomposed patterns to achieve both (i) and (ii). The ***contributions*** are threefold: First, we interpret representation balancing as propensity confusion and derive corresponding theoretical upper bounds for counterfactual error and ITE error based on $\mathcal{H}$-divergence to ensure its rationality; Second, DIGNet transforms the balancing Patterns into Decompositions of Individual propensity confusion and Group distance minimization (*PDIG*) to achieve goal (i), and we empirically find that the PDIG structure learns more effective balancing patterns (task 2) without affecting factual outcome prediction (task 1); Third, DIGNet decomposes representative features into Patterns of Pre-balancing and Balancing Representations (*PPBR*) to achieve goal (ii), and we experimentally confirm that the PPBR approach improves outcome prediction (task 1) without affecting learning balancing patterns (task 2). To better describe our contributions, we illustrate PPBR, PDIG, and the proposed DIGNet in Figure 1. We also illustrate the model sturecture of DIGNet with the variants GNet, INet, DGNet, DINet in Figure 2.

## 1.1 Related Work

The presence of a covariate shift problem stimulates the line of representation balancing works (Johansson et al., 2016; Shalit et al., 2017; Johansson et al., 2022). These works aim to balance the distributions of representations between treated and controlled groups and simultaneously try to maintain representations predictive of factual outcomes. This idea is closely connected with domain adaptation. In particular, the individual treatment effect (ITE) error bound based on Wasserstein distance is similar to the generalization bound in Ben-David et al. (2010); Long et al. (2014); Shen et al. (2018). In addition to Wasserstein distance-based model, our paper derives a new ITE error bound based on $\mathcal{H}$-divergence (Ben-David et al., 2006;

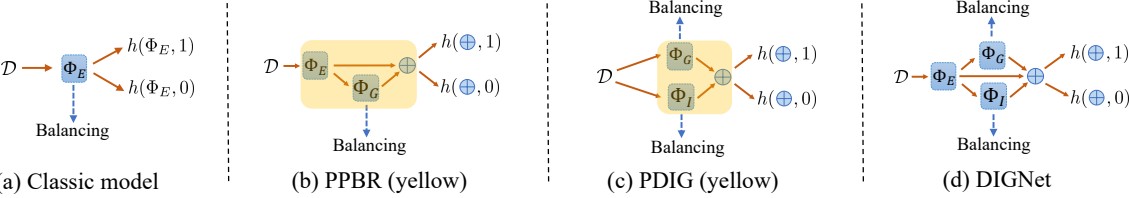

(a) Classic model     (b) PPBR (yellow)     (c) PDIG (yellow)     (d) DIGNet

Figure 1: (a): The classic model (e.g., CFRNet Shalit et al. (2017)) extracts and maps the original data $\mathcal{D}$ into representations $\Phi_E$ to achieve representation balancing over $\Phi_E$, which are referred to as *balancing patterns*. These balanced representations are then used for outcome prediction. (b): The PPBR is represented by the yellow section, where $\Phi_E$ is used for feature extraction and $\Phi_G$ is used for representation balancing, which are termed *pre-balancing patterns* and balancing patterns, respectively. Note that $\Phi_G$ also serves as *decomposed patterns* derived from $\Phi_E$. Subsequently, $\Phi_E$ and $\Phi_G$ are concatenated for predicting outcomes. (c): The PDIG is illustrated as the yellow part, where the balancing patterns consist of two components, $\Phi_G$ and $\Phi_I$, serving for group distance minimization and individual propensity confusion, as detailed in Section 4.2. Afterward, $\Phi_G$ and $\Phi_I$ are concatenated for predicting outcomes. (d): The proposed model, DIGNet, incorporates both PPBR and PDIG. In DIGNet, pre-balancing patterns $\Phi_E$ are decomposed into balancing patterns $\Phi_G$ and $\Phi_I$, which also serve as decomposed patterns. The final outcome prediction is obtained by concatenating $\Phi_E$, $\Phi_G$, and $\Phi_I$.

2010; Ganin et al., 2016). Note that our theoretical results (Section 3.2) and experimental implementations (Section 4.1) differ greatly from Shalit et al. (2017) due to the distinction between Wasserstein distance and $\mathcal{H}$-divergence.

Another recent line of causal representation learning literature investigates efficient neural network structures for treatment effect estimation. Kuang et al. (2017); Hassanpour & Greiner (2019b) extract the original covariates into treatment-specific factors, outcome-specific factors, and confounding factors; X-learner (Künzel et al., 2019) and R-learner (Nie & Wager, 2021) are developed beyond the classic S-learner and T-learner; Curth & van der Schaar (2021) leverage structures for end-to-end learners to counteract the inductive bias towards treatment effect estimation, which is motivated by Makar et al. (2020).

The proposed DIGNet model is built on the PDIG structure and the PPBR approach. The PDIG structure is motivated by multi-task learning, where we design a framework incorporating two specific balancing patterns that share the same pre-balancing patterns. The PPBR approach is motivated by a so-called over-enforcing problem mentioned by Zhang et al. (2020); Assaad et al. (2021); Huang et al. (2022a), where the authors argue that improperly balanced representations can be detrimental predictors for outcome modeling, since such representations can lose the original information that contributes to outcome prediction. Other representation learning methods relevant to treatment effect estimation include Louizos et al. (2017); Yao et al. (2018); Yoon et al. (2018); Shi et al. (2019); Du et al. (2021).

## 2 Preliminaries

**Notations.** Suppose there are the $N$ i.i.d. random variables $\mathcal{D} = \{(\mathbf{X}_i, T_i, Y_i)\}_{i=1}^N$ with observed realizations $\{(\mathbf{x}_i, t_i, y_i)\}_{i=1}^N$, where there are $N_1$ treated units and $N_0$ controlled units. For each unit $i$, $\mathbf{X}_i \in \mathcal{X} \subset \mathbb{R}^d$ denotes $d$-dimensional covariates and $T_i \in \{0,1\}$ denotes the binary treatment, with $e(\mathbf{x}_i) := p(T_i = 1 \mid \mathbf{X}_i = \mathbf{x}_i)$ defined as the propensity score (Rosenbaum & Rubin, 1983). Potential outcome framework (Rubin, 2005) defines the potential outcomes $Y^1, Y^0 \in \mathcal{Y} \subset \mathbb{R}$ for treatment $T = 1$ and $T = 0$, respectively. We let the observed outcome (factual outcome) be $Y = T \cdot Y^1 + (1 - T) \cdot Y^0$, and the unobserved outcome (counterfactual outcome) be $Y = T \cdot Y^0 + (1 - T) \cdot Y^1$. For $t \in \{0,1\}$, let $\tau^t(\mathbf{x}) := \mathbb{E}\left[Y^t \mid \mathbf{X} = \mathbf{x}\right]$ be a function of $Y^t$ w.r.t. $\mathbf{X}$, then our goal is to estimate the individual treatment effect (ITE) $\tau(\mathbf{x}) := \mathbb{E}\left[Y^1 - Y^0 \mid \mathbf{X} = \mathbf{x}\right] = \tau^1(\mathbf{x}) - \tau^0(\mathbf{x})$ [1], and the average treatment effect (ATE)

---

[1]The term $\mathbb{E}\left[Y^1 - Y^0 \mid \mathbf{X} = \mathbf{x}\right]$ is often defined as the Conditional Average Treatment Effect (CATE). In order to maintain consistency with the notion used in the existing causal representation balancing literature, e.g., Shalit et al. (2017), we refer to

$\tau_{ATE} := \mathbb{E}\left[Y^1 - Y^0\right] = \int_{\mathcal{X}} \tau(\mathbf{x})p(\mathbf{x})d\mathbf{x}$. We refer to the representations as patterns. The proposed components PPBR and PDIG are illustrated in Figure 1, and the necessary representation function $\Phi_E$, $\Phi_G$ and $\Phi_I$ are illustrated in Figure 2.

## 2.1 Problem setup

In causal representation balancing works, we denote representation space by $\mathcal{R} \subset \mathbb{R}^d$, and $\Phi : \mathcal{X} \to \mathcal{R}$ is assumed to be a twice-differentiable, one-to-one and invertible function with its inverse $\Psi : \mathcal{R} \to \mathcal{X}$ such that $\Psi(\Phi(\mathbf{x})) = \mathbf{x}$. The densities of the treated and controlled covariates are denoted by $p_{\mathbf{x}}^{T=1} = p^{T=1}(\mathbf{x}) := p(\mathbf{x} \mid T = 1)$ and $p_{\mathbf{x}}^{T=0} = p^{T=0}(\mathbf{x}) := p(\mathbf{x} \mid T = 0)$, respectively. Correspondingly, the densities of the treated and controlled covariates in the representation space are denoted by $p_{\Phi}^{T=1} = p_{\Phi}^{T=1}(\mathbf{r}) := p_{\Phi}(\mathbf{r} \mid T = 1)$ and $p_{\Phi}^{T=0} = p_{\Phi}^{T=0}(\mathbf{r}) := p_{\Phi}(\mathbf{r} \mid T = 0)$, respectively.

Our study is based on the potential outcome framework (Rubin, 2005). Assumption 1 states standard and necessary assumptions to ensure treatment effects are identifiable. Before proceeding with theoretical analysis, we also present some necessary terms and definitions in Definition 1.

**Assumption 1** (Consistency, Overlap, and Unconfoundedness). *Consistency: If the treatment is $t$, then the observed outcome equals $Y^t$. Overlap: The propensity score is bounded away from 0 to 1: $0 < e(\mathbf{x}) < 1$. Unconfoundedness: $Y^t \perp\!\!\!\perp T \mid \mathbf{X}, \forall t \in \{0, 1\}$.*

**Definition 1.** *Let $h : \mathcal{R} \times \{0, 1\} \to \mathcal{Y}$ be an hypothesis defined over the representation space $\mathcal{R}$ such that $h(\Phi(\mathbf{x}), t)$ estimates $y^t$, and $L : \mathcal{Y} \times \mathcal{Y} \to \mathbb{R}_+$ be a loss function (e.g., $L(y, y') = (y - y')^2$). If we define the expected loss for $(\mathbf{x}, t)$ as $\ell_{h,\Phi}(\mathbf{x}, t) = \int_{\mathcal{Y}} L(y^t, h(\Phi(\mathbf{x}), t))p(y^t|\mathbf{x})dy^t$, we then have factual and counterfactual losses, as well as them on the treated and controlled:*

$$\epsilon_F(h, \Phi) = \int_{\mathcal{X} \times \{0,1\}} \ell_{h,\Phi}(\mathbf{x}, t)p(\mathbf{x}, t)d\mathbf{x}dt, \qquad \epsilon_{CF}(h, \Phi) = \int_{\mathcal{X} \times \{0,1\}} \ell_{h,\Phi}(\mathbf{x}, t)p(\mathbf{x}, 1 - t)d\mathbf{x}dt,$$

$$\epsilon_F^{T=1}(h, \Phi) = \int_{\mathcal{X}} \ell_{h,\Phi}(\mathbf{x}, 1)p^{T=1}(\mathbf{x})d\mathbf{x}, \qquad \epsilon_F^{T=0}(h, \Phi) = \int_{\mathcal{X}} \ell_{h,\Phi}(\mathbf{x}, 0)p^{T=0}(\mathbf{x})d\mathbf{x},$$

$$\epsilon_{CF}^{T=1}(h, \Phi) = \int_{\mathcal{X}} \ell_{h,\Phi}(\mathbf{x}, 1)p^{T=0}(\mathbf{x})d\mathbf{x}, \qquad \epsilon_{CF}^{T=0}(h, \Phi) = \int_{\mathcal{X}} \ell_{h,\Phi}(\mathbf{x}, 0)p^{T=1}(\mathbf{x})d\mathbf{x}.$$

If we let $f(\mathbf{x}, t)$ be $h(\Phi(\mathbf{x}), t)$, where $f : \mathcal{X} \times \{0, 1\} \to \mathcal{Y}$ is a prediction function for outcome, then the estimated ITE over $f$ is defined as $\hat{\tau}_f(\mathbf{x}) := f(\mathbf{x}, 1) - f(\mathbf{x}, 0)$. Finally, a better treatment effect estimation can be reformulated as a smaller error in Precision in the expected Estimation of Heterogeneous Effect (PEHE):

$$\epsilon_{PEHE}(f) = \int_{\mathcal{X}} L(\hat{\tau}_f(\mathbf{x}), \tau(\mathbf{x}))p(\mathbf{x})d\mathbf{x}. \tag{1}$$

Here, $\epsilon_{PEHE}(f)$ can also be denoted by $\epsilon_{PEHE}(h, \Phi)$ if we let $f(\mathbf{x}, t)$ be $h(\Phi(\mathbf{x}), t)$.

# 3 Theoretical Results

In this section, we first prove $\epsilon_{PEHE}$ is bounded by $\epsilon_F$ and $\epsilon_{CF}$ in Lemma 1. Next, we revisit and extend the upper bound (Theorem 1) concerning the group distance minimization guided method in Section 3.1. Section 3.2 further discusses the new concept of propensity confusion and the theoretical results based on the individual propensity confusion guided method (Theorem 2). Proofs and additional theoretical results are deferred to Appendix.

**Lemma 1.** *Let functions $h$ and $\Phi$ be as defined in Definition 1, and $L$ be the squared loss function. Recall that $\tau^t(\mathbf{x}) = \mathbb{E}\left[Y^t \mid \mathbf{X} = \mathbf{x}\right]$. Defining $\sigma_y^2 = \min\{\sigma_{y^t}^2(p(\mathbf{x}, t)), \sigma_{y^t}^2(p(\mathbf{x}, 1 - t))\} \ \forall t \in \{0, 1\}$, where $\sigma_{y^t}^2(p(\mathbf{x}, t)) = \int_{\mathcal{X} \times \{0,1\} \times \mathcal{Y}}(y^t - \tau^t(\mathbf{x}))^2 p(y^t|\mathbf{x})p(\mathbf{x}, t)dy^t d\mathbf{x}dt$, we have*

$$\epsilon_{PEHE}(h, \Phi) \le 2(\epsilon_{CF}(h, \Phi) + \epsilon_F(h, \Phi) - 2\sigma_y^2).$$

---

this term as ITE throughout this paper. Note that the original definition of ITE for the $i$-th individual is commonly expressed as the difference between their potential outcomes, represented as $Y_i^1 - Y_i^0$.

Note that similar results will hold as long as $L$ takes forms that satisfy the triangle inequality, so $L$ is not limited to the squared loss. For instance, we give the result for absolute loss in Lemma 6 in Appendix. This extends the result shown in Shalit et al. (2017) that $L$ only takes the squared loss.

## 3.1 GNet: Group Distance Minimization Guided Representation Balancing

Previous causal learning models commonly adopt the group distance minimization guided approach to seek representation balancing via minimizing the distance measured by the Integral Probability Metric (IPM) defined in Definition 2.

**Definition 2.** *Let $\mathcal{G}$ be a function family consisting of functions $g : \mathcal{S} \to \mathbb{R}$. For a pair of distributions $p_1$, $p_2$ over $\mathcal{S}$, the Integral Probability Metric is defined as*

$$IPM_{\mathcal{G}}(p_1, p_2) := \sup_{g \in \mathcal{G}} |\int_{\mathcal{S}} g(s)(p_1(s) - p_2(s))ds|.$$

If $\mathcal{G}$ is the family of 1-Lipschitz functions, we can obtain the so-called 1-Wasserstein distance, denoted by $Wass(p_1, p_2)$ (Sriperumbudur et al., 2012). Next, we present the bounds for counterfactual error $\epsilon_{CF}$ and ITE error $\epsilon_{PEHE}$ using Wasserstein distance in Theorem 1.

**Theorem 1.** *Let $\Phi : \mathcal{X} \to \mathcal{R}$ be an invertible representation with $\Psi$ being its inverse. Define $\sigma_y^2 = \min\{\sigma_{y^t}^2(p(\mathbf{x}, t)), \sigma_{y^t}^2(p(\mathbf{x}, 1 - t))\}$ and $A_y = \max\{A_{y^t}(p(\mathbf{x}, t)), A_{y^t}(p(\mathbf{x}, 1 - t))\}$ $\forall t \in \{0, 1\}$, where $\sigma_{y^t}^2(p(\mathbf{x}, t)) = \int_{\mathcal{X} \times \{0,1\} \times \mathcal{Y}} (y^t - \tau^t(\mathbf{x}))^2 p(y^t|\mathbf{x}) p(\mathbf{x}, t) dy^t d\mathbf{x} dt$ and $A_{y^t}(p(\mathbf{x}, t)) = \int_{\mathcal{X} \times \{0,1\} \times \mathcal{Y}} |y^t - \tau^t(\mathbf{x})| p(y^t|\mathbf{x}) p(\mathbf{x}, t) dy^t d\mathbf{x} dt$ $\forall t \in \{0, 1\}$. Let $p_{\Phi}^{T=1}(\mathbf{r})$, $p_{\Phi}^{T=0}(\mathbf{r})$ be as defined before, $h : \mathcal{R} \times \{0, 1\} \to \mathcal{Y}$, $u := Pr(T = 1)$ and $\mathcal{G}$ be the family of 1-Lipschitz functions. Assume there exists a constant $B_{\Phi} \geq 0$, such that for $t \in \{0, 1\}$, the function $g_{\Phi,h}(\mathbf{r}, t) := \frac{1}{B_{\Phi}} \cdot \ell_{h,\Phi}(\Psi(\mathbf{r}), t) \in \mathcal{G}$. If $L$ is a loss function that satisfies the triangle inequality, we have*

$$\epsilon_{CF}(h, \Phi) \leq (1 - u) \cdot \epsilon_F^{T=1}(h, \Phi) + u \cdot \epsilon_F^{T=0}(h, \Phi) + B_{\Phi} \cdot Wass(p_{\Phi}^{T=1}, p_{\Phi}^{T=0}). \tag{2}$$

*Let loss function $L$ be the squared loss such that $L(y_1, y_2) = (y_1 - y_2)^2$. Then we have:*

$$\epsilon_{PEHE}(h, \Phi) \leq 2(\epsilon_F^{T=1}(h, \Phi) + \epsilon_F^{T=0}(h, \Phi) + B_{\Phi} \cdot Wass(p_{\Phi}^{T=1}, p_{\Phi}^{T=0}) - 2\sigma_y^2). \tag{3}$$

*Let loss function $L$ be the absolute loss such that $L(y_1, y_2) = |y_1 - y_2|$. Then we have:*

$$\epsilon_{PEHE}(h, \Phi) \leq \epsilon_F^{T=1}(h, \Phi) + \epsilon_F^{T=0}(h, \Phi) + B_{\Phi} \cdot Wass(p_{\Phi}^{T=1}, p_{\Phi}^{T=0}) + 2A_y. \tag{4}$$

Our Theorem 1 is a more general result compared to previous literature since it holds for any $L$ that takes forms satisfying the triangle inequality. For instance, Theorem 1 will reduce to the results in Shalit et al. (2017) if $L$ is the squared loss as shown in equation 3. We also give the result for absolute loss in Theorem 1 as shown in equation 4. We refer to a model built on Theorem 1 as **GNet** (aka CFR-Wass in Shalit et al. (2017)) since it is based on group distance minimization.

## 3.2 INet: Individual Propensity Confusion Guided Representation Balancing

The propensity score plays a central role to treatment effect estimation because it characterizes the probability that one receives treatment (Rosenbaum & Rubin, 1983). Unlike previous studies that primarily employ propensity scores for matching or weighting, we emphasize that the propensity score can also serve as a natural indicator of whether representations are adequately balanced. Specifically, when it becomes challenging to distinguish whether each unit in the representation space is treated or controlled, the representations are believed adequately balanced. Consequently, the concept of representation balancing can be intuitively understood as propensity confusion, which provides a logical interpretation and justification for minimizing the $\mathcal{H}$-divergence-based error bound for ITE. In the following content, we will first establish the upper bounds for counterfactual error and ITE error utilizing the $\mathcal{H}$-divergence, as stated in Theorem 2.

**Definition 3.** *Given a pair of distributions $p_1$, $p_2$ over $\mathcal{S}$, and a hypothesis binary function class $\mathcal{H}$, the $\mathcal{H}$-divergence between $p_1$ and $p_2$ is defined as*

$$d_{\mathcal{H}}(p_1, p_2) := 2 \sup_{\eta \in \mathcal{H}} |Pr_{p_1}[\eta(s) = 1] - Pr_{p_2}[\eta(s) = 1]|. \tag{5}$$

**Theorem 2.** *Let $\Phi : \mathcal{X} \to \mathcal{R}$ be an invertible representation with $\Psi$ being its inverse. Define $\sigma_y^2 = \min\{\sigma_{y^t}^2(p(\mathbf{x}, t)), \sigma_{y^t}^2(p(\mathbf{x}, 1 - t))\}$ and $A_y = \max\{A_{y^t}(p(\mathbf{x}, t)), A_{y^t}(p(\mathbf{x}, 1 - t))\}$ $\forall t \in \{0, 1\}$, where $\sigma_{y^t}^2(p(\mathbf{x}, t)) = \int_{\mathcal{X} \times \{0,1\} \times \mathcal{Y}} (y^t - \tau^t(\mathbf{x}))^2 p(y^t | \mathbf{x}) p(\mathbf{x}, t) dy^t d\mathbf{x} dt$ and $A_{y^t}(p(\mathbf{x}, t)) = \int_{\mathcal{X} \times \{0,1\} \times \mathcal{Y}} |y^t - \tau^t(\mathbf{x})| p(y^t | \mathbf{x}) p(\mathbf{x}, t) dy^t d\mathbf{x} dt$ $\forall t \in \{0, 1\}$. Let $p_{\Phi}^{T=1}(\mathbf{r})$, $p_{\Phi}^{T=0}(\mathbf{r})$ be as defined before, $h : \mathcal{R} \times \{0, 1\} \to \mathcal{Y}$, $u := Pr(T = 1)$ and $\mathcal{H}$ be the family of binary functions. Assume that there exists a constant $K \geq 0$ such that $\int_{\mathcal{Y}} L(y, y') dy \leq K$ $\forall y' \in \mathcal{Y}$. If $L$ is a loss function that satisfies the triangle inequality, we have*

$$\epsilon_{CF}(h, \Phi) \leq (1 - u) \cdot \epsilon_F^{T=1}(h, \Phi) + u \cdot \epsilon_F^{T=0}(h, \Phi) + \frac{K}{2} d_{\mathcal{H}}(p_{\Phi}^{T=1}, p_{\Phi}^{T=0}). \tag{6}$$

*Let loss function $L$ be the squared loss such that $L(y_1, y_2) = (y_1 - y_2)^2$. Then we have:*

$$\epsilon_{PEHE}(h, \Phi) \leq 2(\epsilon_F^{T=1}(h, \Phi) + \epsilon_F^{T=0}(h, \Phi) + \frac{K}{2} d_{\mathcal{H}}(p_{\Phi}^{T=1}, p_{\Phi}^{T=0}) - 2\sigma_y^2). \tag{7}$$

*Let loss function $L$ be the absolute loss such that $L(y_1, y_2) = |y_1 - y_2|$. Then we have:*

$$\epsilon_{PEHE}(h, \Phi) \leq \epsilon_F^{T=1}(h, \Phi) + \epsilon_F^{T=0}(h, \Phi) + \frac{K}{2} d_{\mathcal{H}}(p_{\Phi}^{T=1}, p_{\Phi}^{T=0}) + 2A_y. \tag{8}$$

The details of the proof of Theorem 2 are given in Theorem 2 in Appendix. Theorem 2 holds for forms of $L$ as long as $L$ takes forms that satisfy the triangle inequality. For example, we give the result for squared loss in equation 7, and the result for absolute loss in equation 8. Note that detailed proofs of Theorem 2 in Appendix suggest that our theoretical derivations are not trivial extensions from other $\mathcal{H}$-divergence related works (Ben-David et al., 2006; 2010; Ganin et al., 2016) and causal representation balancing works (Shalit et al., 2017; Johansson et al., 2022), which is also one of our main theoretical contributions. We refer to a model built on Theorem 2 as **INet** since it is based on individual propensity confusion.

## 4 Method

In the preceding section, we presented theoretical results that guarantee the rationale behind representation balancing methods relying on Wasserstein distance and $\mathcal{H}$-divergence. Moving on to Section 4.1, we will begin by revisiting GNet, which can be considered as CFRNet Shalit et al. (2017), a Wasserstein distance-based representation balancing network. Additionally, we will demonstrate how Theorem 2 can be connected with propensity confusion, leading us to introduce INet, a $\mathcal{H}$-divergence-based representation balancing network. Subsequently, in Section 4.2, we will introduce the PDIG and PPBR components for representation balancing within the scheme of decomposed patterns. Building upon GNet, INet, PPBR, and PDIG, we will design DIGNet, a novel representation balancing network with decomposed patterns.

### 4.1 Representation Balancing without decomposed Patterns

In representation balancing models, given the input data tuples $(\mathbf{x}, \mathbf{t}, \mathbf{y}) = \{(\mathbf{x}_i, t_i, y_i)\}_{i=1}^N$, the original covariates $\mathbf{x}$ are extracted by some representation function $\Phi(\cdot)$, and representations $\Phi(\mathbf{x})$ are then fed into the outcome functions $h^1(\cdot) := h(\cdot, 1)$ and $h^0(\cdot) := h(\cdot, 0)$ that estimate the potential outcome $y^1$ and $y^0$, respectively. Finally, the factual outcome can be predicted by $h^t(\cdot) = t h^1(\cdot) + (1 - t) h^0(\cdot)$, and the corresponding outcome loss is

$$\mathcal{L}_y(\mathbf{x}, \mathbf{t}, \mathbf{y}; \Phi, h^t) = \frac{1}{N} \sum_{i=1}^N L(h^t(\Phi(\mathbf{x}_i)), y_i). \tag{9}$$

If models such as GNet and INet do not have decomposition modes, both outcome prediction and representation balancing will rely on the extracted features $\Phi(\mathbf{x})$. Below we will introduce the objectives of GNet and INet.

**Objective of GNet.** GNet learns the balancing patterns over $\Phi$ by minimizing the group distance loss $\mathcal{L}_G(\mathbf{x}, \mathbf{t}; \Phi) = Wass(\{\Phi(\mathbf{x}_i)\}_{i:t_i=0}, \{\Phi(\mathbf{x}_i)\}_{i:t_i=1})$. If the original covariates $\mathbf{x}$ are extracted by the feature extractor $\Phi_E(\cdot)$, then the final objective of GNet is

$$\min_{\Phi_E, h^t} \quad \mathcal{L}_y(\mathbf{x}, \mathbf{t}, \mathbf{y}; \Phi_E, h^t) + \alpha_1 \mathcal{L}_G(\mathbf{x}, \mathbf{t}; \Phi_E). \tag{10}$$

For the convenience of the reader, we illustrate the structure of GNet in Figure 2(a).

**Objective of INet.** Next, we detail how Theorem 2 is related to propensity confusion and give the objective of INet. Let $\mathbb{I}(a)$ be an indicator function that gives 1 if $a$ is true, and $\mathcal{H}$ be the family of binary functions as defined in Theorem 2. The representation balancing seeks to minimize empirical $\mathcal{H}$-divergence $\hat{d}_{\mathcal{H}}(p_{\Phi}^{T=1}, p_{\Phi}^{T=0})$ such that

$$\hat{d}_{\mathcal{H}}(p_{\Phi}^{T=1}, p_{\Phi}^{T=0}) = 2 \left( 1 - \min_{\eta \in \mathcal{H}} \left[ \frac{1}{N} \sum_{i:\eta(\Phi(\mathbf{x}_i))=0} \mathbb{I}[t_i = 1] + \frac{1}{N} \sum_{i:\eta(\Phi(\mathbf{x}_i))=1} \mathbb{I}[t_i = 0] \right] \right). \tag{11}$$

The "min" part in equation 11 indicates that the optimal classifier $\eta^* \in \mathcal{H}$ minimizes the classification error between the estimated treatment $\eta^*(\Phi(\mathbf{x}_i))$ and the observed treatment $t_i$, i.e., discriminating whether $\Phi(\mathbf{x}_i)$ is controlled ($T = 0$) or treated ($T = 1$). As a result, $\hat{d}_{\mathcal{H}}(p_{\Phi}^{T=1}, p_{\Phi}^{T=0})$ will be large if $\eta^*$ can easily distinguish whether $\Phi(\mathbf{x}_i)$ is treated or controlled, i.e., the optimal classification error is small. In contrast, $\hat{d}_{\mathcal{H}}(p_{\Phi}^{T=1}, p_{\Phi}^{T=0})$ will be small if it is hard for $\eta^*$ to determine whether $\Phi(\mathbf{x}_i)$ is treated or controlled, i.e., the optimal classification error is large. Therefore, the prerequisite of a small $\mathcal{H}$-divergence is to find a map $\Phi$ such that any classifier $\eta \in \mathcal{H}$ will get confused about the probability of $\Phi(\mathbf{x}_i)$ being treated or controlled. To achieve this goal, we define a discriminator $\pi(\mathbf{r}) : \mathcal{R} \to [0, 1]$ that estimates the propensity score of $\mathbf{r}$, which can be regarded as a surrogate for $\eta(\mathbf{r})$. The classification error for the $i^{th}$ individual can be empirically approximated by the cross-entropy loss between $\pi(\Phi(\mathbf{x}_i))$ and $t_i$:

$$\mathcal{L}_t(t_i, \pi(\Phi(\mathbf{x}_i))) = - \left[ t_i \log \pi(\Phi(\mathbf{x}_i)) + (1 - t_i) \log(1 - \pi(\Phi(\mathbf{x}_i))) \right]. \tag{12}$$

As a consequence, we aim to find an optimal discriminator $\pi^*$ for equation 11 such that $\pi^*$ maximizes the probability that treatment is correctly classified of the total population:

$$\max_{\pi \in \mathcal{H}} \mathcal{L}_I(\mathbf{x}, \mathbf{t}; \Phi, \pi) = \max_{\pi \in \mathcal{H}} \left[ -\frac{1}{N} \sum_{i=1}^{N} \mathcal{L}_t(t_i, \pi(\Phi(\mathbf{x}_i))) \right]. \tag{13}$$

Given the feature extractor $\Phi_E(\cdot)$, the objective of INet can be formulated as a min-max game:

$$\min_{\Phi_E, h^t} \max_{\pi} \quad \mathcal{L}_y(\mathbf{x}, \mathbf{t}, \mathbf{y}; \Phi_E, h^t) + \alpha_2 \mathcal{L}_I(\mathbf{x}, \mathbf{t}; \Phi_E, \pi). \tag{14}$$

As stated in equation 14, INet achieves the representation balancing through a min-max formulation, following a strategy of empirical approximation of $\mathcal{H}$-divergence in Ganin et al. (2016). In the maximization, the discriminator $\pi$ is trained to maximize the probability that treatment is correctly classified. This forces $\pi(\Phi_E(\mathbf{x}_i))$ closer to the true propensity score $e(\mathbf{x}_i)$. In the minimization, the feature extractor $\Phi_E$ is trained to fool the discriminator $\pi$. This confuses $\pi$ such that $\pi(\Phi_E(\mathbf{x}_i))$ cannot correctly specify the true propensity score $e(\mathbf{x}_i)$. Eventually, the representations are balanced as it is difficult for $\pi$ to determine the propensity of $\Phi(\mathbf{x}_i)$ being treated or controlled. For the convenience of the reader, we illustrate the structure of INet in Figure 2(b).

### 4.2 Representation Balancing with Decomposed Patterns

**PDIG.** Previous demonstrations have shown that GNet is thriving and widely adopted, while INet is meaningful and interpretable. Nevertheless, they still face the trade-off between representation balancing and outcome modeling. To this end, we expect to capture more effective balancing patterns by turning the

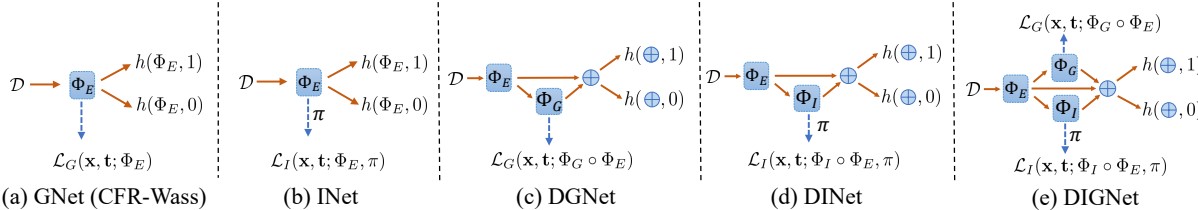

Figure 2: Illustrations of the network architecture of the five models studied in Section 5.

balancing Patterns into Decompositions of Individual propensity confusion and Group distance minimization (PDIG). More specifically, the covariates $\mathbf{x}$ are extracted by the feature extractor $\Phi_E(\cdot)$, and then $\Phi_E(\mathbf{x})$ are fed into the balancing networks $\Phi_G(\cdot)$ and $\Phi_I(\cdot)$ for group distance minimization and individual propensity confusion, respectively. Finally, the losses for the two separate balancing patterns are

$$\min_{\Phi_G} \ \mathcal{L}_G(\mathbf{x}, \mathbf{t}; \Phi_G \circ \Phi_E),$$
$$\min_{\Phi_I} \max_{\pi} \ \mathcal{L}_I(\mathbf{x}, \mathbf{t}; \Phi_I \circ \Phi_E, \pi). \tag{15}$$

Here, $\circ$ denotes the composition of two functions, indicating that $\Phi(\cdot)$ in $\mathcal{L}_G(\mathbf{x}, \mathbf{t}; \Phi)$ and $\mathcal{L}_I(\mathbf{x}, \mathbf{t}; \Phi, \pi)$ are replaced by $\Phi_G(\Phi_E(\cdot))$ and $\Phi_I(\Phi_E(\cdot))$, respectively.

**PPBR.** Motivated by the discussion in Section 1, we aim to design a framework that is capable of capturing Patterns of Pre-balancing and Balancing Representations (PPBR) to improve outcome modeling. To this end, the representation balancing patterns $\Phi_G(\Phi_E(\mathbf{x}))$ and $\Phi_I(\Phi_E(\mathbf{x}))$ are first learned over $\Phi_G$ and $\Phi_I$, while $\Phi_E$ is remained fixed as pre-balancing patterns. Furthermore, we concatenate the balancing representations $\Phi_G(\Phi_E(\mathbf{x}))$ and $\Phi_I(\Phi_E(\mathbf{x}))$ with the pre-balancing representations $\Phi_E(\mathbf{x})$ as attributes for outcome prediction. As a result, the proxy features used for outcome predictions are $\Phi_E(\mathbf{x}) \oplus \Phi_G(\Phi_E(\mathbf{x})) \oplus \Phi_I(\Phi_E(\mathbf{x}))$, where $\oplus$ indicates the concatenation by column. For example, if $\mathbf{a} = [1, 2]$ and $\mathbf{b} = [3, 4]$, then $\mathbf{a} \oplus \mathbf{b} = [1, 2, 3, 4]$.

**Objective of DIGNet.** Combining with PDIG and PPBR, we propose a new model architecture, DIGNet, as illustrated in Figure 2(e). The objective of DIGNet is separated into four stages:

$$\min_{\Phi_G} \ \alpha_1 \mathcal{L}_G(\mathbf{x}, \mathbf{t}; \Phi_G \circ \Phi_E), \tag{16}$$

$$\max_{\pi} \ \alpha_2 \mathcal{L}_I(\mathbf{x}, \mathbf{t}; \Phi_I \circ \Phi_E, \pi), \tag{17}$$

$$\min_{\Phi_I} \ \alpha_2 \mathcal{L}_I(\mathbf{x}, \mathbf{t}; \Phi_I \circ \Phi_E, \pi), \tag{18}$$

$$\min_{\Phi_E, \Phi_I, \Phi_G, h^t} \ \mathcal{L}_y(\mathbf{x}, \mathbf{t}, \mathbf{y}; \Phi_E \oplus (\Phi_I \circ \Phi_E) \oplus (\Phi_G \circ \Phi_E), h^t). \tag{19}$$

Within each iteration, DIGNet manages to minimize the group distance via equation 16, and plays an adversarial game to achieve propensity confusion through equation 17 and equation 18. In equation 19, DIGNet updates both the pre-balancing and balancing patterns $\Phi_E, \Phi_I, \Phi_G$ along with the outcome function $h^t$ to minimize the outcome prediction loss.

**DGNet and DINet.** For further ablation studies, we also designed two models, DGNet and DINet. The two models can be considered as either DIGNet without PDIG, or GNet and INet with PPBR. The structures of DGNet and DINet are shown in Figure 2(c) and Figure 2(d), and the objectives of DGNet and DINet are deferred to Section A.6 in Appendix.

## 5 Experiments

In non-randomized observational data, the ground truth regarding treatment effects remains inaccessible due to the absence of counterfactual information. Therefore, we use simulated data and semi-synthetic

Figure 3: T-SNE visualizations of the covariates as $\gamma$ varies. Red represents the treatment group and blue represents the control group. A larger $\gamma$ indicates a greater imbalance between the two groups.

benchmark data to test the performance of our methods and other baseline models. In this section, our primary focus revolves around addressing two key questions:

**Q1.** Is PDIG helpful in ITE estimation by learning more effective balancing patterns without affecting factual outcome prediction? In other words, can DIGNet which incorporates the PDIG structure achieve superior ITE estimation performances with more balanced representations compared to DGNet and DINet?

**Q2.** Is PPBR helpful in ITE estimation by improving factual outcome prediction without affecting learning balancing patterns? In other words, can DGNet and DINet that involve PPBR enhance ITE estimation performances with smaller factual outcome errors compared to standard representation balancing models such as GNet and INet?

### 5.1 Experimental Settings

**Simulation data.** Previous causal inference works assess the model effectiveness by varying the distribution imbalance of covariates in treated and controlled groups at different levels Yao et al. (2018); Yoon et al. (2018); Du et al. (2021). As suggested in Assaad et al. (2021), we draw 1000 observational data points from the following data generating strategy:

$$\mathbf{X}_i \sim \mathcal{N}(\mathbf{0}, \sigma^2 \cdot [\rho \mathbf{1}_p \mathbf{1}_p' + (1-\rho)\mathbf{I}_p]),$$
$$T_i \mid \mathbf{X}_i \sim \text{Bernoulli}(1/(1 + \exp(-\gamma \mathbf{X}_i))),$$
$$Y_i^0 = \boldsymbol{\beta_0'}\mathbf{X}_i + \xi_i, \quad Y_i^1 = \boldsymbol{\beta_1'}\mathbf{X}_i + \xi_i, \quad \xi_i \sim \mathcal{N}(0,1).$$

Here, $\mathbf{1}_p$ denotes the $p$-dimensional all-ones vector and $\mathbf{I}_p$ denotes the identity matrix of size $p$. We fix $p = 10, \rho = 0.3, \sigma^2 = 2, \boldsymbol{\beta_0'} = [0.3, ..., 0.3], \boldsymbol{\beta_1'} = [1.3, ..., 1.3]$ and vary $\gamma \in \{0.25, 0.5, 0.75, 1, 1.5, 2, 3\}$ to yield different levels of selection bias. As seen in Figure 3, selection bias becomes more severe with $\gamma$ increasing. For each $\gamma$, we repeat the above data generating process to generate 30 different datasets, with each dataset split by the ratio of 56%/24%/20% as training/validation/test sets.

**Semi-synthetic data.** The IHDP dataset is introduced by Hill (2011). This dataset consists of 747 samples with 25-dimensional covariates collected from real-world randomized experiments. Selection bias is created by removing some of treated samples. The goal is to estimate the effect of special visits (treatment) on cognitive scores (outcome). The potential outcomes are generated using the NPCI package Dorie (2021). We use the same 1000 datasets as used in Shalit et al. (2017), with each dataset split by the ratio of 63%/27%/10% as training/validation/test sets.

**Models and metrics.** In simulation experiments, we perform comprehensive comparisons between INet, GNet, DINet, DGNet, and DIGNet in terms of the mean and standard error for the following metrics: $\sqrt{\epsilon_{PEHE}}$, $\sqrt{\epsilon_{CF}}$, and $\sqrt{\epsilon_F}$ with $L$ defined in Definition 1 being the squared loss, as well as the empirical approximations of $Wass(p_\Phi^{T=1}, p_\Phi^{T=0})$ and $d_{\mathcal{H}}(p_\Phi^{T=1}, p_\Phi^{T=0})$ (denoted by $Wass$ and $\hat{d}_{\mathcal{H}}$, respectively). Note that as shown in Figure 2, $Wass$ is over $\Phi_E$ for GNet while over $\Phi_G$ for DGNet and DIGNet; $\hat{d}_{\mathcal{H}}$ is over $\Phi_E$ for INet while over $\Phi_I$ for DINet and DIGNet. To analyze the source of gain in simulation studies, we fairly compare models by ensuring that each model shares the same hyperparameters, e.g., learning rate, the number of layers and units for $(\Phi_E, \Phi_G, \Phi_I, f^t)$, and $(\alpha_1, \alpha_2)$. Note that we apply an early stopping

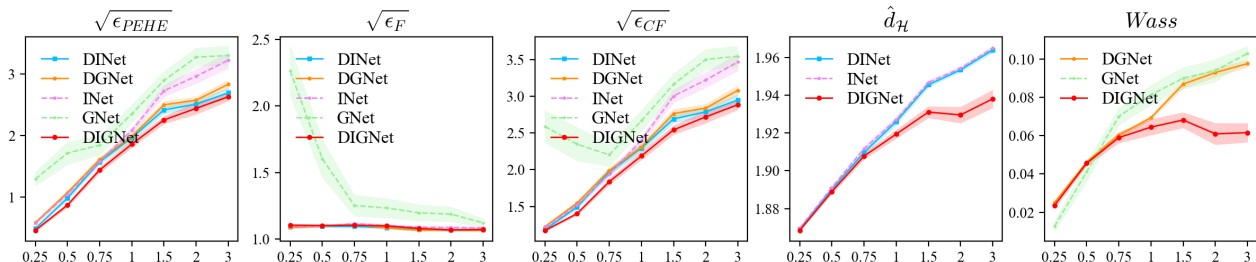

Figure 4: Plots of model performances on test set for different metrics as $\gamma$ varies in $\{0.25, 0.5, 0.75, 1, 1.5, 2, 3\}$. Each graph shows the average of 30 runs with standard errors shaded.

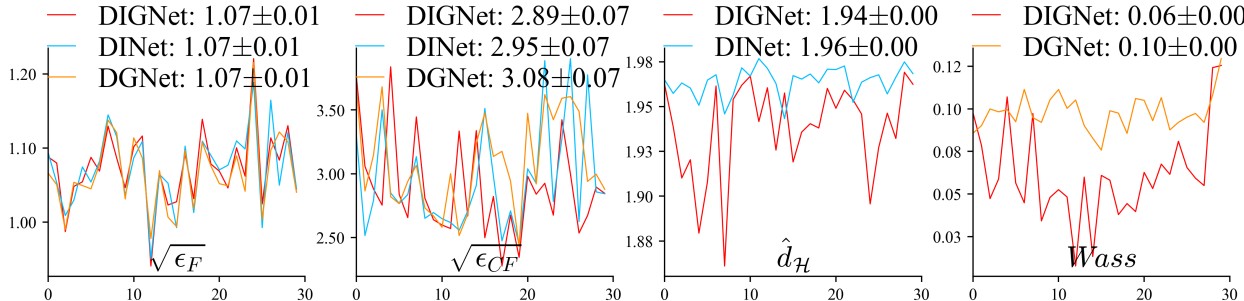

Figure 5: Plots of model performances on test set for $\sqrt{\epsilon_F}$, $\sqrt{\epsilon_{CF}}$, $\hat{d}_\mathcal{H}$, and $Wass$ when $\gamma = 3$. Each graph plots the metric for 30 runs. Mean $\pm$ std of each metric averaged across 30 runs are reported on the top.

rule to all models as Shalit et al. (2017) do. In IHDP experiment, we use $\sqrt{\epsilon_{PEHE}}$, as well as an additional metric $\epsilon_{ATE} = |\hat{\tau}_{ATE} - \tau_{ATE}|$ to evaluate performances of various causal models (see them in Table 3). More descriptions of the implementation details, as well as the analysis of training time and training stability, are detailed in Section A.5 of Appendix.

**Device.** All the experiments are run on Dell 7920 with one 16-core Intel Xeon Gold 6250 3.90GHz CPU and three NVIDIA Quadro RTX 6000 GPUs.

### 5.2 Results and Analysis

**Varying selection bias.** We first make a general comparison between models with the degree of covariate imbalance increasing, and the relevant results are shown in Figure 4. There are four main observations:

1. DIGNet attains the lowest $\sqrt{\epsilon_{PEHE}}$ across all datasets, while GNet have inferior performances than other models;

2. DINet and DGNet outperform INet and GNet regarding $\sqrt{\epsilon_{CF}}$ and $\sqrt{\epsilon_{PEHE}}$;

3. INet, DINet, and DGNet perform similarly to DIGNet on factual outcome estimations ($\sqrt{\epsilon_F}$), but cannot compete with DIGNet in terms of counterfactual estimations ($\sqrt{\epsilon_{CF}}$);

4. DIGNet achieves smaller $\hat{d}_\mathcal{H}$ (or $Wass$) than DINet and INet (or DGNet and GNet), especially when the covariate shift problem is severe (e.g., when $\gamma > 1$).

In conclusion, the above study has produced several noteworthy findings. Firstly, finding (1) reveals that our proposed DIGNet model consistently performs well in ITE estimation. Secondly, as indicated by finding (2), implementing the PPBR approach can enhance the predictive accuracy of factual and counterfactual outcomes. Lastly, findings (3) and (4) highlight the role of PDIG structure in promoting the simultaneous reinforcement and complementarity of group distance minimization and individual propensity confusion.

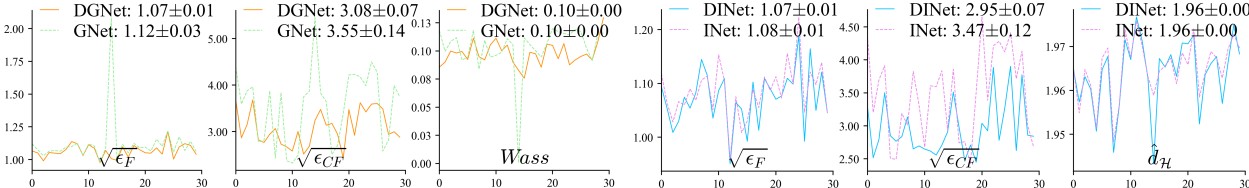

Figure 6: Plots of model performances on test set for different metrics when $\gamma = 3$. Each graph plots the metric for 30 runs, with mean $\pm$ std averaged across 30 runs reported on the top.

Table 1: Training- & test- set $\sqrt{\epsilon_{PEHE}}$ & $\epsilon_{ATE}$ when $\gamma = 3$. Mean $\pm$ standard error of 30 runs.

|  | Training set | | Test set | |
|---|---|---|---|---|
|  | $\sqrt{\epsilon_{PEHE}}$ | $\epsilon_{ATE}$ | $\sqrt{\epsilon_{PEHE}}$ | $\epsilon_{ATE}$ |
| GNet | 3.30±0.15 | 2.58±0.14 | 3.30±0.16 | 2.59±0.14 |
| INet | 3.24±0.11 | 2.46±0.09 | 3.22±0.12 | 2.47±0.10 |
| DGNet | 2.86±0.06 | 2.15±0.03 | 2.83±0.07 | 2.15±0.04 |
| DINet | 2.70±0.06 | 2.12±0.04 | 2.69±0.08 | 2.13±0.05 |
| DIGNet | 2.66±0.07 | 2.04±0.05 | 2.63±0.07 | 2.03±0.04 |

Table 2: Training- & test- set $\sqrt{\epsilon_{PEHE}}$ & $\epsilon_{ATE}$ on IHDP. Mean $\pm$ standard error of 100 runs.

|  | Training set | | Test set | |
|---|---|---|---|---|
|  | $\sqrt{\epsilon_{PEHE}}$ | $\epsilon_{ATE}$ | $\sqrt{\epsilon_{PEHE}}$ | $\epsilon_{ATE}$ |
| GNet | 0.71±0.15 | 0.12±0.01 | 0.77±0.18 | 0.15±0.02 |
| INet | 0.66±0.09 | 0.13±0.01 | 0.72±0.11 | 0.15±0.02 |
| DGNet | 0.53±0.07 | 0.11±0.01 | 0.60±0.09 | 0.13±0.01 |
| DINet | 0.57±0.12 | 0.13±0.01 | 0.60±0.11 | 0.14±0.01 |
| DIGNet | 0.42±0.02 | 0.11±0.01 | 0.45±0.04 | 0.12±0.01 |

Moving forward, our subsequent analysis will step further into understanding the effectiveness of our proposed methods, building upon these preliminary conclusions.

**Source of gain.** To further investigate the above findings, we choose the case with high selection bias ($\gamma = 3$) to explore the source of gain for PDIG and PPBR. We report model performances (mean $\pm$ std) averaged over 30 training and test sets in Table 1 and plot specific metrics of 30 runs on test set in Figure 5 and Figure 6. Below we discuss the source of gain in detail.

(1) Ablation study for PDIG: The PDIG structure is manifest to be effective in capturing more balanced patterns, without affecting factual outcome predictions. As depicted in Figure 4, DIGNet exhibits more balanced representations, irrespective of whether the discrepancy is measured by $\hat{d}_{\mathcal{H}}$ or $Wass$, while DIGNet, DINet, and DGNet demonstrate comparable estimates of factual outcomes ($\sqrt{\epsilon_F}$). In particular, by comparing DIGNet with DGNet and DINet in Figure 5, we find that the PDIG structure does not affect the factual outcome estimation ($\sqrt{\epsilon_F}$). Nevertheless, DIGNet achieves smaller $\hat{d}_{\mathcal{H}}$ with a $|1.94/1.96 - 1| = 1.0\%$ reduction (or $Wass$ with a $|0.06/0.10 - 1| = 40\%$ reduction) compared with DINet (or DGNet). This indicates that PDIG enables group distance minimization and individual propensity confusion to complement and reinforce each other, thereby learning better balancing patterns. This advantage translates into superior counterfactual estimation, with DIGNet reduceing $\sqrt{\epsilon_{CF}}$ by $|2.89/2.95-1| = 2.0\%$ and $|2.89/3.08-1| = 6.2\%$ compared to DINet and DGNet, respectively. Consequently, due to the effective capture of balancing patterns by PDIG, DIGNet shows superiority in treatment effect estimation ($\sqrt{\epsilon_{PEHE}}$ and $\epsilon_{ATE}$) compared to DGNet and DINet, as demonstrated in Table 1.

(2) Ablation study for PPBR: The PPBR approach plays an essential role in improving outcome predictions, without affecting learning balancing patterns. From Figure 6, we gain an important insight that the difference in learned representation balancing patterns, measured by $Wass$ (or $\hat{d}_{\mathcal{H}}$), between DGNet and GNet (or DINet and INet), is negligible. This implies that PPBR does not impact the representation balancing task. However, PPBR can improve the predictive power of factual outcomes, resulting in a $|1.07/1.12 - 1| = 4.5\%$ reduction in $\sqrt{\epsilon_F}$ for GNet and a $|1.07/1.08 - 1| = 0.9\%$ reduction for INet. Notably, this improvement is particularly pronounced in counterfactual estimation, where $\sqrt{\epsilon_{CF}}$ is reduced by $|3.08/3.55-1| = 13.2\%$ for GNet and $|2.95/3.47 - 1| = 15.0\%$ for INet. Benefiting from the advantage of PPBR, the treatment effect errors ($\sqrt{\epsilon_{PEHE}}$ and $\epsilon_{ATE}$) attained by DINet and DGNet are significantly smaller than those attained by INet and GNet, as shown in Table 1.

**Comparisons on IHDP benchmark.** We first conduct an ablation study for PDIG and PPBR on 1-100 IHDP datasets and report the results in Table 2. Further, we undergo comparisons between DIGNet and other causal models on 1-1000 IHDP datasets and report the results in Table 3. Note that "-" indicates either the result is not reproducible or the original paper does not report relevant values. Table 2 shows that DINet and DGNet are superior to INet and GNet but inferior to DIGNet concerning treatment effect estimation, suggesting that each component of PDIG and PPBR is advantageous for treatment effect estimation. For example, on the test set, DINet reduces $\sqrt{\epsilon_{PEHE}}$ by $|0.60/0.72 - 1| = 16.7\%$ for INet, and DIGNet achieves $|0.45/0.60 - 1| = 25\%$ error reduction regarding $\sqrt{\epsilon_{PEHE}}$ for DINet. This is consistent with the findings before: PPBR and PDIG are beneficial to treatment effect estimation. Table 3 demonstrates that models that involve either propensity score or representation balancing (e.g., DKLITE, CFR-X, BWCFR-X, and MBRL) attain $\sqrt{\epsilon_{PEHE}}$ and $\epsilon_{ATE}$ of $0.57 \sim 0.70$ and $0.13 \sim 0.19$, respectively. Compared to the second-best method, DIGNet improves performance by $|0.45/0.57 - 1| = 21\%$ and $|0.12/0.13 - 1| = 7.7\%$ regarding $\sqrt{\epsilon_{PEHE}}$ and $\epsilon_{ATE}$, respectively, revealing the prominent outperformance of the proposed method. Moreover, it is noticeable that DIGNet achieves the lowest errors overwhelmingly across datasets and metrics, indicating that the proposed method has the most robust performance.

## 6 Conclusion

In this paper, we derive a theoretical ITE bound based on $\mathcal{H}$-divergence and connect representation balancing with the concept of propensity confusion. More importantly, we propose the components of PDIG and PPBR, on which we construct a decomposition network structure DIGNet for treatment effect estimation. Comprehensive experiments verify that PDIG and PPBR follow different pathways to improve the generalization of counterfactual and ITE estimation, and such improvements are confirmed to be very significant. In particular, PDIG helps the model capture more effective representation balancing patterns without affecting outcome prediction, while PPBR preserves patterns predictive of outcomes to enhance the outcome prediction without influencing learning balancing patterns. We sincerely hope that our findings can constitute an important step to inspire more research concerning the generalization of representation balancing models in counterfactual estimation.

Table 3: Training- & test- set $\sqrt{\epsilon_{PEHE}}$ & $\epsilon_{ATE}$ on IHDP. Mean $\pm$ standard error of 1000 runs.

| | Training set | | Test set | |
|---|---|---|---|---|
| | $\sqrt{\epsilon_{PEHE}}$ | $\epsilon_{ATE}$ | $\sqrt{\epsilon_{PEHE}}$ | $\epsilon_{ATE}$ |
| OLS/LR$_1$ (Johansson et al., 2016) | $5.8 \pm .3$ | $.73 \pm .04$ | $5.8 \pm .3$ | $.94 \pm .06$ |
| OLS/LR$_2$ (Johansson et al., 2016) | $2.4 \pm .1$ | $.14 \pm .01$ | $2.5 \pm .1$ | $.31 \pm .02$ |
| k-NN (Crump et al., 2008) | $2.1 \pm .1$ | $.14 \pm .01$ | $4.1 \pm .2$ | $.79 \pm .05$ |
| BART (Chipman et al., 2010) | $2.1 \pm .1$ | $.23 \pm .01$ | $2.3 \pm .1$ | $.34 \pm .02$ |
| CF (Wager & Athey, 2018) | $3.8 \pm .2$ | $.18 \pm .01$ | $3.8 \pm .2$ | $.40 \pm .03$ |
| CEVAE (Louizos et al., 2017) | $2.7 \pm .1$ | $.34 \pm .01$ | $2.6 \pm .1$ | $.46 \pm .02$ |
| SITE (Yao et al., 2018) | $.69 \pm .0$ | $.22 \pm .01$ | $.75 \pm .0$ | $.24 \pm .01$ |
| GANITE (Yoon et al., 2018) | $1.9 \pm .4$ | $.43 \pm .05$ | $2.4 \pm .4$ | $.49 \pm .05$ |
| BLR (Johansson et al., 2016) | $5.8 \pm .3$ | $.72 \pm .04$ | $5.8 \pm .3$ | $.93 \pm .05$ |
| BNN (Johansson et al., 2016) | $2.2 \pm .1$ | $.37 \pm .03$ | $2.1 \pm .1$ | $.42 \pm .03$ |
| TARNet (Shalit et al., 2017) | $.88 \pm .0$ | $.26 \pm .01$ | $.95 \pm .0$ | $.28 \pm .01$ |
| CFR-Wass (GNet) (Shalit et al., 2017) | $.73 \pm .0$ | $.12 \pm .01$ | $.81 \pm .0$ | $.15 \pm .01$ |
| Dragonnet (Shi et al., 2019) | $1.3 \pm .4$ | $.14 \pm .01$ | $1.3 \pm .5$ | $.20 \pm .05$ |
| DKLITE (Zhang et al., 2020) | $.52 \pm .0$ | $-$ | $.65 \pm .03$ | $-$ |
| CFR-ISW (Hassanpour & Greiner, 2019a) | $-$ | $-$ | $.70 \pm .0$ | $.19 \pm .03$ |
| BWCFR-OW (Assaad et al., 2021) | $-$ | $-$ | $.65 \pm .0$ | $.18 \pm .01$ |
| BWCFR-MW (Assaad et al., 2021) | $-$ | $-$ | $.63 \pm .0$ | $.19 \pm .01$ |
| BWCFR-TruncIPW (Assaad et al., 2021) | $-$ | $-$ | $.63 \pm .0$ | $.19 \pm .01$ |
| MBRL (Huang et al., 2022a) | $.52 \pm .0$ | $.12 \pm .01$ | $.57 \pm .0$ | $.13 \pm .01$ |
| DIGNet (Ours) | $\mathbf{.42 \pm .0}$ | $\mathbf{.11 \pm .01}$ | $\mathbf{.45 \pm .0}$ | $\mathbf{.12 \pm .01}$ |

Our paper primarily offers effective empirical solutions to address the trade-off challenge between domain invariance and domain discrimination given the absence of a well-established theoretical solution in the current research field. Therefore, it is also important to step beyond these empirical insights into future theoretical studies aimed at resolving the trade-off problem. A promising avenue for future theoretical investigations involves the development of new distributional divergences or theoretical upper bounds that can robustly handle the trade-off issue. Exploring the integration of distributionally robust optimization and nonparametric inference methods in this context would be intriguing. Furthermore, deriving an upper bound with an analytically optimal solution to the trade-off problem could be valuable, albeit possibly requiring additional assumptions. Empirical studies can focus on discouraging the redundancy of shared information within the PDIG structure and improving the optimization efficacy of DIGNet's objective.

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

# A   Appendix

## A.1   Preliminaries

We start by making some assumptions about the distribution we concern, and give some necessary definitions and results. We make the strong ignorability assumption, which assume that there exists a joint distribution $p(\mathbf{X}, T, Y^0, Y^1)$ such that conditioning on covariate $\mathbf{X}$, the potential outcomes $Y^0$, $Y^1$ are independent of $T$, i.e., $(Y^0, Y^1) \perp\!\!\!\perp T \mid \mathbf{X}$, and the propensity score $e(\mathbf{x}) := p(T = 1 \mid \mathbf{X} = \mathbf{x})$ is bounded away from 0 to 1, i.e.

$0 < e(\mathbf{x}) < 1$. Recall that we also assume consistency, i.e., if the treatment is $t$, the observed outcome equals $Y^t$. These assumptions are crucial conditions that make individual treatment effect identifiable (Imbens & Wooldridge, 2009).

**Definition 1.** *The individual treatment effect (ITE) for unit $\mathbf{x}$ is:*

$$\tau(\mathbf{x}) = \mathbb{E}\left[Y^1 - Y^0 \mid \mathbf{X} = \mathbf{x}\right].$$

Let $\tau^t(\mathbf{x}) := \mathbb{E}\left[Y^t \mid \mathbf{X} = \mathbf{x}\right]$, we have $\tau(\mathbf{x}) = \tau^1(\mathbf{x}) - \tau^0(\mathbf{x})$. Let $f : \mathcal{X} \times \{0,1\} \to \mathcal{Y}$ be a prediction function.

**Definition 2.** *The individual treatment effect estimate can be defined as:*

$$\hat{\tau}_f(\mathbf{x}) := f(\mathbf{x}, 1) - f(\mathbf{x}, 0).$$

**Definition 3.** *(Hill, 2011) Let $L : \mathcal{Y} \times \mathcal{Y} \to \mathbb{R}_+$ be a loss function. The expected Precision in Estimation of Heterogeneous Effect (PEHE) loss of $f$ is:*

$$\epsilon_{PEHE}(f) = \int_{\mathcal{X}} L(\hat{\tau}_f(\mathbf{x}), \tau(\mathbf{x})) p(\mathbf{x}) d\mathbf{x}.$$

**Definition 4.** *The covariates' distributions in the treated and controlled groups can be denoted by $p^{T=1}(\mathbf{x}) := p(\mathbf{x} \mid T = 1)$ and $p^{T=0}(\mathbf{x}) := p(\mathbf{x} \mid T = 0)$, respectively.*

In our causal representation balancing approach, we assume that the representation function $\Phi : \mathcal{X} \to \mathcal{R}$ is a twice-differentiable, one-to-one function, where $\mathcal{R} \subset \mathbb{R}^d$ is the representation space. Then, we can denote $\Psi : \mathcal{R} \to \mathcal{X}$ by the inverse of $\Phi$ and the induced distribution of $\mathbf{r}$ by $p_\Phi$.

**Definition 5.** *The covariates' distributions in the treated and controlled groups over $\mathcal{R}$ can be denoted by $p_\Phi^{T=1}(\mathbf{r}) := p_\Phi(\mathbf{r} \mid T = 1)$ and $p_\Phi^{T=0}(\mathbf{r}) := p_\Phi(\mathbf{r} \mid T = 0)$, respectively.*

Let $h : \mathcal{R} \times \{0,1\} \to \mathcal{Y}$ be an hypothesis defined over the representation space $\mathcal{R}$, such that $f(\mathbf{x}, t) = h(\Phi(\mathbf{x}), t)$.

**Definition 6.** *The expected loss for the unit and treatment pair $(\mathbf{x}, t)$ is :*

$$\ell_{h,\Phi}(\mathbf{x}, t) = \int_{\mathcal{Y}} L(y^t, h(\Phi(\mathbf{x}), t)) p(y^t | \mathbf{x}) dy^t.$$

**Definition 7.** *The expected factual loss and counterfactual losses of $h$ and $\Phi$ are, respectively:*

$$\epsilon_F(h, \Phi) = \int_{\mathcal{X} \times \{0,1\}} \ell_{h,\Phi}(\mathbf{x}, t) p(\mathbf{x}, t) d\mathbf{x} dt,$$

$$\epsilon_{CF}(h, \Phi) = \int_{\mathcal{X} \times \{0,1\}} \ell_{h,\Phi}(\mathbf{x}, t) p(\mathbf{x}, 1 - t) d\mathbf{x} dt.$$

**Definition 8.** *The expected treated and control losses are:*

$$\epsilon_F^{T=1}(h, \Phi) = \int_{\mathcal{X}} \ell_{h,\Phi}(\mathbf{x}, 1) p^{T=1}(\mathbf{x}) d\mathbf{x},$$

$$\epsilon_F^{T=0}(h, \Phi) = \int_{\mathcal{X}} \ell_{h,\Phi}(\mathbf{x}, 0) p^{T=0}(\mathbf{x}) d\mathbf{x},$$

$$\epsilon_{CF}^{T=1}(h, \Phi) = \int_{\mathcal{X}} \ell_{h,\Phi}(\mathbf{x}, 1) p^{T=0}(\mathbf{x}) d\mathbf{x},$$

$$\epsilon_{CF}^{T=0}(h, \Phi) = \int_{\mathcal{X}} \ell_{h,\Phi}(\mathbf{x}, 0) p^{T=1}(\mathbf{x}) d\mathbf{x}.$$

Let $u := Pr(T = 1)$ be the proportion of treated in the population. We then have the result:

**Lemma 1.**

$$\epsilon_F(h, \Phi) = u \cdot \epsilon_F^{T=1}(h, \Phi) + (1 - u) \cdot \epsilon_F^{T=0}(h, \Phi),$$

$$\epsilon_{CF}(h, \Phi) = (1 - u) \cdot \epsilon_{CF}^{T=1}(h, \Phi) + u \cdot \epsilon_{CF}^{T=0}(h, \Phi).$$

Noting that $p(\mathbf{x}, t) = u \cdot p^{T=1}(\mathbf{x}) + (1 - u) \cdot p^{T=0}(\mathbf{x})$, the results can be easily obtained from the Definitions 7 and 8. This Lemma follows the results in Shalit et al. (2017).

**Definition 9.** *Let $\mathcal{G}$ be a function family consisting of functions $g : \mathcal{S} \to \mathbb{R}$. For a pair of distributions $p_1$, $p_2$ over $\mathcal{S}$, define the Integral Probability Metric:*

$$IPM_{\mathcal{G}}(p_1, p_2) = \sup_{g \in \mathcal{G}} |\int_{\mathcal{S}} g(s)(p_1(s) - p_2(s))ds|.$$

Let $\mathcal{G}$ be the family of 1-Lipschitz functions, we obtain the so-called 1-Wasserstein distance between distributions, which we denote $Wass(p_1, p_2)$ (Sriperumbudur et al., 2012).

**Definition 10.** *Given a pair of distributions $p_1$, $p_2$ over $\mathcal{S}$, and a hypothesis binary function class $\mathcal{H}$, the $\mathcal{H}$-divergence between $p_1$ and $p_2$ is*

$$d_{\mathcal{H}}(p_1, p_2) = 2 sup_{\eta \in \mathcal{H}} |Pr_{p_1}[\eta(s) = 1] - Pr_{p_2}[\eta(s) = 1]|.$$

**Lemma 2.** *Let $\mathcal{G}$ in Definition 9 be the family of binary functions. Then we obtain* $\sup_{\eta \in \mathcal{H}} |\int_{\mathcal{S}} \eta(s)(p_1(s) - p_2(s))ds| = \frac{1}{2} d_{\mathcal{H}}(p_1, p_2).$

*Proof.* Let $\mathbb{I}(\cdot)$ denotes an indicator function.

$$d_{\mathcal{H}}(p_1, p_2)$$

$$= 2 \sup_{\eta \in \mathcal{H}} \left| \int_{\eta(s)=1} (p_1(s) - p_2(s))ds \right|$$

$$= 2 \sup_{\eta \in \mathcal{H}} \left| \int_{\mathcal{S}} \mathbb{I}(\eta(s) = 1)(p_1(s) - p_2(s))ds \right|$$

$$= 2 \sup_{\eta \in \mathcal{H}} \left| \int_{\mathcal{S}} \eta(s)(p_1(s) - p_2(s))ds \right| \tag{20}$$

The last equation is because an indicator function is also a binary function. $\square$

### A.2   Bounds for conterfactual error $\epsilon_{CF}$

We first derive the counterfactual error bounds when using Wasserstein distance. The following Lemma 3 and corresponding proof is identical to the Lemma 1 in (Shalit et al., 2017).

**Lemma 3.** *Let $\Phi : \mathcal{X} \to \mathcal{R}$ be an invertible representation with $\Psi$ being its inverse. Let $p_{\Phi}^{T=1}(\mathbf{r})$, $p_{\Phi}^{T=0}(\mathbf{r})$ be as defined before. Let $h : \mathcal{R} \times \{0, 1\} \to \mathcal{Y}$, $u := Pr(T = 1)$ and $\mathcal{G}$ be the family of 1-Lipschitz functions. Assume there exists a constant $B_{\Phi} \geq 0$, such that for $t = 0, 1$, the function $g_{\Phi, h}(\mathbf{r}, t) := \frac{1}{B_{\Phi}} \cdot \ell_{h, \Phi}(\Psi(\mathbf{r}), t) \in \mathcal{G}$. Then we have:*

$$\epsilon_{CF}(h, \Phi) \leq (1 - u) \cdot \epsilon_F^{T=1}(h, \Phi) + u \cdot \epsilon_F^{T=0}(h, \Phi) + B_{\Phi} \cdot Wass(p_{\Phi}^{T=1}, p_{\Phi}^{T=0}).$$

*Proof.*

$$\epsilon_{CF}(h, \Phi) - [(1-u) \cdot \epsilon_F^{T=1}(h, \Phi) + u \cdot \epsilon_F^{T=0}(h, \Phi)]$$
$$= [(1-u) \cdot \epsilon_{CF}^{T=1}(h, \Phi) + u \cdot \epsilon_{CF}^{T=0}(h, \Phi)] - [(1-u) \cdot \epsilon_F^{T=1}(h, \Phi) + u \cdot \epsilon_F^{T=0}(h, \Phi)]$$
$$= (1-u) \cdot [\epsilon_{CF}^{T=1}(h, \Phi) - \epsilon_F^{T=1}(h, \Phi)] + u \cdot [\epsilon_{CF}^{T=0}(h, \Phi) - \epsilon_F^{T=0}(h, \Phi)]$$
$$= (1-u) \int_{\mathcal{X}} \ell_{h,\Phi}(\mathbf{x}, 1)(p^{T=0}(\mathbf{x}) - p^{T=1}(\mathbf{x}))d\mathbf{x} + u \int_{\mathcal{X}} \ell_{h,\Phi}(\mathbf{x}, 0)(p^{T=1}(\mathbf{x}) - p^{T=0}(\mathbf{x}))d\mathbf{x} \qquad (21)$$
$$= (1-u) \int_{\mathcal{R}} \ell_{h,\Phi}(\Psi(\mathbf{r}), 1)(p_\Phi^{T=0}(\mathbf{r}) - p_\Phi^{T=1}(\mathbf{r}))d\mathbf{r} + u \int_{\mathcal{R}} \ell_{h,\Phi}(\Psi(\mathbf{r}), 0)(p_\Phi^{T=1}(\mathbf{r}) - p_\Phi^{T=0}(\mathbf{r}))d\mathbf{r} \qquad (22)$$
$$= B_\Phi \cdot (1-u) \int_{\mathcal{R}} \frac{1}{B_\Phi} \ell_{h,\Phi}(\Psi(\mathbf{r}), 1)(p_\Phi^{T=0}(\mathbf{r}) - p_\Phi^{T=1}(\mathbf{r}))d\mathbf{r}$$
$$+ B_\Phi \cdot u \int_{\mathcal{R}} \frac{1}{B_\Phi} \ell_{h,\Phi}(\Psi(\mathbf{r}), 0)(p_\Phi^{T=1}(\mathbf{r}) - p_\Phi^{T=0}(\mathbf{r}))d\mathbf{r}$$
$$\leq B_\Phi \cdot (1-u) \sup_{g \in \mathcal{G}} |\int_{\mathcal{R}} g(\mathbf{r})(p_\Phi^{T=0}(\mathbf{r}) - p_\Phi^{T=1}(\mathbf{r}))d\mathbf{r}|$$
$$+ B_\Phi \cdot u \cdot \sup_{g \in \mathcal{G}} |\int_{\mathcal{R}} g(\mathbf{r})(p_\Phi^{T=1}(\mathbf{r}) - p_\Phi^{T=0}(\mathbf{r}))d\mathbf{r}| \qquad (23)$$
$$= B_\Phi \cdot Wass(p_\Phi^{T=1}, p_\Phi^{T=0}) \qquad (24)$$

Equation (21) is by Definition 8; equation (22) is by the change of formula, $p_\Phi^{T=0}(\mathbf{r}) = p^{T=0}(\Psi(\mathbf{r}))J_\Psi(\mathbf{r})$, $p_\Phi^{T=1}(\mathbf{r}) = p^{T=1}(\Psi(\mathbf{r}))J_\Psi(\mathbf{r})$, where $J_\Psi(\mathbf{r})$ is the absolute of the determinant of the Jacobian of $\Psi(\mathbf{r})$; inequality (23) is by the premise that $\frac{1}{B_\Phi} \cdot \ell_{h,\Phi}(\Psi(\mathbf{r}), t) \in \mathcal{G}$ for $t = 0, 1$, and (24) is by Definition 9 of an IPM. $\qquad \square$

The crucial condition in Lemma 3 is that $g_{\Phi,h}(\mathbf{r}, t) := \frac{1}{B_\Phi} \cdot \ell_{h,\Phi}(\Psi(\mathbf{r}), t) \in \mathcal{G}$. Bounds for $B_\Phi$ can be given to evaluate this constant when under more assumptions about the loss function $L$, the Lipschitz constants of $p(y^t|\mathbf{x})$, $h$, and the condition number of the Jacobian of $\Phi$. These assumptions and the specific bounds for $B_\Phi$ can be seen in supplement Section A.3 of (Shalit et al., 2017).

Now we turn to derive the counterfactual error bounds for the $\mathcal{H}$-divergence case.

**Assumption 1.** *There exists a constant $K > 0$ such that $\sup\limits_{y_2 \in \mathcal{Y}, \mathbf{x} \in \mathcal{X}} \int_{\mathcal{Y}} L(y_1, y_2)p(y_1|\mathbf{x})dy_1 \leq K$.*

**Lemma 4.** *Let $\Phi : \mathcal{X} \to \mathcal{R}$ be an invertible representation with $\Psi$ being its inverse. Let $p_\Phi^{T=1}(\mathbf{r})$, $p_\Phi^{T=0}(\mathbf{r})$ be as defined before. Let $h : \mathcal{R} \times \{0, 1\} \to \mathcal{Y}$, $u := Pr(T = 1)$ and $\mathcal{H}$ be the family of binary functions. Assume loss function $L$ obeys the Assumption 1. Then we have:*

$$\epsilon_{CF}(h, \Phi) \leq (1-u) \cdot \epsilon_F^{T=1}(h, \Phi) + u \cdot \epsilon_F^{T=0}(h, \Phi) + \frac{K}{2}d_{\mathcal{H}}(p_\Phi^{T=1}, p_\Phi^{T=0}).$$

*Proof.*

$$\epsilon_{CF}(h, \Phi) - [(1-u) \cdot \epsilon_F^{T=1}(h, \Phi) + u \cdot \epsilon_F^{T=0}(h, \Phi)]$$

$$= (1-u) \int_{\mathcal{R}} \ell_{h,\Phi}(\Psi(\mathbf{r}), 1)(p_\Phi^{T=0}(\mathbf{r}) - p_\Phi^{T=1}(\mathbf{r}))d\mathbf{r} + u \int_{\mathcal{R}} \ell_{h,\Phi}(\Psi(\mathbf{r}), 0)(p_\Phi^{T=1}(\mathbf{r}) - p_\Phi^{T=0}(\mathbf{r}))d\mathbf{r} \qquad (25)$$

$$\leq (1-u) \int_{p_\Phi^{T=0} > p_\Phi^{T=1}} \ell_{h,\Phi}(\Psi(\mathbf{r}), 1)(p_\Phi^{T=0}(\mathbf{r}) - p_\Phi^{T=1}(\mathbf{r}))d\mathbf{r}$$

$$+ u \int_{p_\Phi^{T=1} > p_\Phi^{T=0}} \ell_{h,\Phi}(\Psi(\mathbf{r}), 0)(p_\Phi^{T=1}(\mathbf{r}) - p_\Phi^{T=0}(\mathbf{r}))d\mathbf{r} \qquad (26)$$

$$\leq (1-u)K \int_{p_\Phi^{T=0} > p_\Phi^{T=1}} (p_\Phi^{T=0}(\mathbf{r}) - p_\Phi^{T=1}(\mathbf{r}))d\mathbf{r} + u \cdot K \int_{p_\Phi^{T=1} > p_\Phi^{T=0}} (p_\Phi^{T=1}(\mathbf{r}) - p_\Phi^{T=0}(\mathbf{r}))d\mathbf{r} \qquad (27)$$

$$= (1-u)K \int_{\mathcal{R}} \mathbb{I}(p_\Phi^{t=0} > p_\Phi^{T=1})(p_\Phi^{T=0}(\mathbf{r}) - p_\Phi^{T=1}(\mathbf{r}))d\mathbf{r}$$

$$+ u \cdot K \int_{\mathcal{R}} \mathbb{I}(p_\Phi^{T=1} > p_\Phi^{T=0})(p_\Phi^{T=1}(\mathbf{r}) - p_\Phi^{T=0}(\mathbf{r}))d\mathbf{r}$$

$$\leq (1-u)K \sup_{\eta \in \mathcal{H}} | \int_{\mathcal{R}} \eta(\mathbf{r})(p_\Phi^{T=1}(\mathbf{r}) - p_\Phi^{T=0}(\mathbf{r}))d\mathbf{r}|$$

$$+ u \cdot K \cdot \sup_{\eta \in \mathcal{H}} | \int_{\mathcal{R}} \eta(\mathbf{r})(p_\Phi^{T=1}(\mathbf{r}) - p_\Phi^{T=0}(\mathbf{r}))d\mathbf{r}| \qquad (28)$$

$$\leq K \cdot \sup_{\eta \in \mathcal{H}} | \int_{\mathcal{R}} \eta(\mathbf{r})((p_\Phi^{T=1}(\mathbf{r}) - p_\Phi^{T=0}(\mathbf{r})))d\mathbf{r}|$$

$$= \frac{K}{2} d_{\mathcal{H}}(p_\Phi^{T=1}, p_\Phi^{T=0}) \qquad (29)$$

Equation (25) is same to equation (22); equation (26) is by $\ell_{h,\Phi} \geq 0$ for all $\mathbf{r}$ and $t$; inequality (27) is by Definition 6 and Assumption 1; inequality (28) is because an indicator function is also a binary function; equation (29) is by (20) in Lemma 2. $\square$

### A.3 Bounds for the PEHE loss $\epsilon_{PEHE}$

We first state two lemmas for $\epsilon_{PEHE}$ with respect to two different loss functions: the squared loss and the absolute loss. In fact, similar lemmas hold for loss functions that satisfy the (relaxed) triangle inequalities.

**Definition 11.** *The expected variance of $y^t$ with regard to $p(\mathbf{x}, t)$ is:*

$$\sigma_{y^t}^2(p(\mathbf{x}, t)) = \int_{\mathcal{X} \times \{0,1\} \times \mathcal{Y}} (y^t - \tau^t(\mathbf{x}))^2 p(y^t|\mathbf{x})p(\mathbf{x}, t)dy^t d\mathbf{x}dt,$$

*and define:*

$$\sigma_y^2 = \min\{\sigma_{y^t}^2(p(\mathbf{x}, t)), \sigma_{y^t}^2(p(\mathbf{x}, 1-t))\}.$$

**Lemma 5.** *Let loss function $L$ be the squared loss, $L(y_1, y_2) = (y_1 - y_2)^2$. For any function $f : \mathcal{X} \times \{0,1\} \to \mathcal{Y}$, and distribution $p(\mathbf{x}, t)$ over $\mathcal{X} \times \{0,1\}$,* we have the same results in Shalit et al. (2017):

$$\epsilon_{PEHE}(h, \Phi) \leq 2(\epsilon_{CF}(h, \Phi) + \epsilon_F(h, \Phi) - 2\sigma_y^2)$$

*Proof.* We denote $\epsilon_{PEHE}(f) = \epsilon_{PEHE}(h, \Phi)$, $\epsilon_F(f) = \epsilon_F(h, \Phi)$, $\epsilon_{CF}(f) = \epsilon_{CF}(h, \Phi)$ for $f(\mathbf{x}, t) = h(\Phi(\mathbf{x}), t)$.

$$\epsilon_{PEHE}(f)$$

$$= \int_{\mathcal{X}} ((f(\mathbf{x}, 1) - f(\mathbf{x}, 0)) - (\tau^1(\mathbf{x}) - \tau^0(\mathbf{x})))^2 p(\mathbf{x}) d\mathbf{x}$$

$$\leq 2 \int_{\mathcal{X}} ((f(\mathbf{x}, 1) - \tau^1(\mathbf{x}))^2 + (f(\mathbf{x}, 0) - \tau^0(\mathbf{x}))^2) p(\mathbf{x}) d\mathbf{x} \tag{30}$$

$$= 2 \int_{\mathcal{X}} (f(\mathbf{x}, 1) - \tau^1(\mathbf{x}))^2 p(\mathbf{x}, T = 1) d\mathbf{x} + 2 \int_{\mathcal{X}} (f(\mathbf{x}, 0) - \tau^0(\mathbf{x}))^2 p(\mathbf{x}, T = 0) d\mathbf{x}$$

$$+ 2 \int_{\mathcal{X}} (f(\mathbf{x}, 1) - \tau^1(\mathbf{x}))^2 p(\mathbf{x}, T = 0) d\mathbf{x} + 2 \int_{\mathcal{X}} (f(\mathbf{x}, 0) - \tau^0(\mathbf{x}))^2 p(\mathbf{x}, T = 1) d\mathbf{x} \tag{31}$$

$$= 2 \int_{\mathcal{X} \times \{0,1\}} (f(\mathbf{x}, t) - \tau^t(\mathbf{x}))^2 p(\mathbf{x}, t) d\mathbf{x} dt + 2 \int_{\mathcal{X} \times \{0,1\}} (f(\mathbf{x}, t) - \tau^t(\mathbf{x}))^2 p(\mathbf{x}, 1 - t) d\mathbf{x} dt.$$

Inequality (30) is because the relaxed triangle inequality, $(x + y)^2 \leq 2(x^2 + y^2)$; equation (31) is because $p(\mathbf{x}) = p(\mathbf{x}, T = 0) + p(\mathbf{x}, T = 1)$.

$$\epsilon_F(f)$$

$$= \int_{\mathcal{X} \times \{0,1\} \times \mathcal{Y}} (f(\mathbf{x}, t) - y^t)^2 p(y^t|\mathbf{x}) p(\mathbf{x}, t) dy^t d\mathbf{x} dt$$

$$= \int_{\mathcal{X} \times \{0,1\} \times \mathcal{Y}} (f(\mathbf{x}, t) - \tau^t(\mathbf{x}))^2 p(y^t|\mathbf{x}) p(\mathbf{x}, t) dy^t d\mathbf{x} dt$$

$$+ \int_{\mathcal{X} \times \{0,1\} \times \mathcal{Y}} (\tau^t(\mathbf{x}) - y^t)^2 p(y^t|\mathbf{x}) p(\mathbf{x}, t) dy^t d\mathbf{x} dt$$

$$+ 2 \int_{\mathcal{X} \times \{0,1\} \times \mathcal{Y}} (f(\mathbf{x}, t) - \tau^t(\mathbf{x}))(\tau^t(\mathbf{x}) - y^t) p(y^t|\mathbf{x}) p(\mathbf{x}, t) dy^t d\mathbf{x} dt \tag{32}$$

$$= \int_{\mathcal{X} \times \{0,1\}} (f(\mathbf{x}, t) - \tau^t(\mathbf{x}))^2 p(\mathbf{x}, t) d\mathbf{x} dt + \sigma_{y^t}^2(p(\mathbf{x}, t)) \tag{33}$$

Equation (33) is by Definition 11 and last term in equation (32) equals to zero, since $\tau^t(\mathbf{x}) = \int_{\mathcal{Y}} y^t p(y^t|\mathbf{x}) dy_t$. A similar result can be obtained for $\epsilon_{CF}$:

$$\epsilon_{CF}(f) = \int_{\mathcal{X} \times \{0,1\}} (f(\mathbf{x}, t) - \tau^t(\mathbf{x}))^2 p(\mathbf{x}, 1 - t) d\mathbf{x} dt + \sigma_{y^t}^2(p(\mathbf{x}, 1 - t)).$$

Combining these results and Definition 11, we have

$$\epsilon_{PEHE}(h, \Phi) \leq 2(\epsilon_F(f) - \sigma_{y^t}^2(p(\mathbf{x}, t))) + 2(\epsilon_{CF}(f) - \sigma_{y^t}^2(p(\mathbf{x}, 1 - t)))$$

$$\leq 2(\epsilon_{CF}(h, \Phi) + \epsilon_F(h, \Phi) - 2\sigma_y^2).$$

$\square$

For the absolute loss $L(y_1, y_2) = |y_1 - y_2|$ that satisfies triangle inequality, the upper bound in Lemma 5 will replace the standard deviation $\sigma_y^2$ by mean absolute deviation $A_y$.

**Definition 12.** *The mean absolute deviation of $y^t$ with regard to $p(\mathbf{x}, t)$ is:*

$$A_{y^t}(p(\mathbf{x}, t)) = \int_{\mathcal{X} \times \{0,1\} \times \mathcal{Y}} |y^t - \tau^t(\mathbf{x})| p(y^t|\mathbf{x}) p(\mathbf{x}, t) dy^t d\mathbf{x} dt,$$

*and define:*

$$A_y = \max\{A_{y^t}(p(\mathbf{x}, t)), A_{y^t}(p(\mathbf{x}, 1 - t))\}.$$

**Lemma 6.** *Let loss function $L$ be the absolute loss, $L(y_1, y_2) = |y_1 - y_2|$. For any function $f : \mathcal{X} \times \{0, 1\} \to \mathcal{Y}$, and distribution $p(\mathbf{x}, t)$ over $\mathcal{X} \times \{0, 1\}$:*

$$\epsilon_{PEHE}(h, \Phi) \leq \epsilon_{CF}(h, \Phi) + \epsilon_F(h, \Phi) + 2A_y.$$

*Proof.* Recall that $\epsilon_{PEHE}(f) = \epsilon_{PEHE}(h, \Phi)$, $\epsilon_F(f) = \epsilon_F(h, \Phi)$, $\epsilon_{CF}(f) = \epsilon_{CF}(h, \Phi)$ for $f(\mathbf{x}, t) = h(\Phi(\mathbf{x}), t)$.

$$\epsilon_{PEHE}(f)$$
$$= \int_{\mathcal{X}} |(f(\mathbf{x}, 1) - f(\mathbf{x}, 0)) - (\tau^1(\mathbf{x}) - \tau^0(\mathbf{x}))| p(\mathbf{x}) d\mathbf{x}$$
$$\leq \int_{\mathcal{X}} (|f(\mathbf{x}, 1) - \tau^1(\mathbf{x})| + |f(\mathbf{x}, 0) - \tau^0(\mathbf{x})|) p(\mathbf{x}) d\mathbf{x} \tag{34}$$
$$= \int_{\mathcal{X}} |f(\mathbf{x}, 1) - \tau^1(\mathbf{x})| p(\mathbf{x}, T = 1) d\mathbf{x} + \int_{\mathcal{X}} |f(\mathbf{x}, 0) - \tau^0(\mathbf{x})| p(\mathbf{x}, T = 0) d\mathbf{x}$$
$$+ \int_{\mathcal{X}} |f(\mathbf{x}, 1) - \tau^1(\mathbf{x})| p(\mathbf{x}, T = 0) d\mathbf{x} + \int_{\mathcal{X}} |f(\mathbf{x}, 0) - \tau^0(\mathbf{x})| p(\mathbf{x}, T = 1) d\mathbf{x} \tag{35}$$
$$= \int_{\mathcal{X} \times \{0,1\}} |f(\mathbf{x}, t) - \tau^t(\mathbf{x})| p(\mathbf{x}, t) d\mathbf{x} dt + \int_{\mathcal{X} \times \{0,1\}} |f(\mathbf{x}, t) - \tau^t(\mathbf{x})| p(\mathbf{x}, 1 - t) d\mathbf{x} dt.$$

Inequality (34) is because triangle inequality, $|x + y| \leq |x| + |y|$; equation (35) is because $p(\mathbf{x}) = p(\mathbf{x}, T = 0) + p(\mathbf{x}, T = 1)$.

$$\epsilon_F(f)$$
$$= \int_{\mathcal{X} \times \{0,1\} \times \mathcal{Y}} |f(\mathbf{x}, t) - y^t| p(y^t | \mathbf{x}) p(\mathbf{x}, t) dy^t d\mathbf{x} dt$$
$$\geq \int_{\mathcal{X} \times \{0,1\} \times \mathcal{Y}} |f(\mathbf{x}, t) - \tau^t(\mathbf{x})| p(y^t | \mathbf{x}) p(\mathbf{x}, t) dy^t d\mathbf{x} dt$$
$$- \int_{\mathcal{X} \times \{0,1\} \times \mathcal{Y}} |\tau^t(\mathbf{x}) - y^t| p(y^t | \mathbf{x}) p(\mathbf{x}, t) dy^t d\mathbf{x} dt \tag{36}$$
$$= \int_{\mathcal{X} \times \{0,1\}} |f(\mathbf{x}, t) - \tau^t(\mathbf{x})| p(\mathbf{x}, t) d\mathbf{x} dt - A_{y^t}(p(\mathbf{x}, t)). \tag{37}$$

Inequality (36) is also because $|x + y| \geq |x| - |y|$, equation (37) is by Definition 12. A similar result can be obtained for $\epsilon_{CF}$:

$$\epsilon_{CF}(f) = \int_{\mathcal{X} \times \{0,1\}} |f(\mathbf{x}, t) - \tau^t(\mathbf{x})| p(\mathbf{x}, 1 - t) d\mathbf{x} dt - A_{y^t}(p(\mathbf{x}, 1 - t)).$$

Combining these results and Definition 12, we have

$$\epsilon_{PEHE}(h, \Phi) \leq \epsilon_F(f) + A_{y^t}(p(\mathbf{x}, t)) + \epsilon_{CF}(f) + A_{y^t}(p(\mathbf{x}, 1 - t))$$
$$\leq \epsilon_{CF}(h, \Phi) + \epsilon_F(h, \Phi) + 2A_y.$$

$\square$

We summarize the upper bounds of $\epsilon_{CF}$ and $\epsilon_{PEHE}$ above, and give the final bounds for these two distance using the squared and absolute loss, respectively.

**Theorem 1.** *Let $\Phi : \mathcal{X} \to \mathcal{R}$ be an invertible representation with $\Psi$ being its inverse. Let $p_\Phi^{T=1}(\mathbf{r})$, $p_\Phi^{T=0}(\mathbf{r})$ be as defined before. Let $h : \mathcal{R} \times \{0, 1\} \to \mathcal{Y}$, $u := Pr(T = 1)$ and $\mathcal{G}$ be the family of 1-Lipschitz functions. Assume there exists a constant $B_\Phi \geq 0$, such that for $t = 0, 1$, the function $g_{\Phi,h}(\mathbf{r}, t) := \frac{1}{B_\Phi} \cdot \ell_{h,\Phi}(\Psi(\mathbf{r}), t) \in \mathcal{G}$.*

Let loss function $L$ be the squared loss, $L(y_1, y_2) = (y_1 - y_2)^2$. Then we have *(same results in Shalit et al. (2017))*:

$$\epsilon_{PEHE}(h, \Phi)$$

$$\leq 2(\epsilon_{CF}(h, \Phi) + \epsilon_F(h, \Phi) - 2\sigma_y^2) \tag{38}$$

$$\leq 2(\epsilon_F^{T=1}(h, \Phi) + \epsilon_F^{T=0}(h, \Phi) + B_\Phi \cdot Wass(p_\Phi^{T=1}, p_\Phi^{T=0}) - 2\sigma_y^2) \tag{39}$$

Let loss function $L$ be the absolute loss, $L(y_1, y_2) = |y_1 - y_2|$. Then we have:

$$\epsilon_{PEHE}(h, \Phi)$$

$$\leq \epsilon_{CF}(h, \Phi) + \epsilon_F(h, \Phi) + 2A_y \tag{40}$$

$$\leq \epsilon_F^{T=1}(h, \Phi) + \epsilon_F^{T=0}(h, \Phi) + B_\Phi \cdot Wass(p_\Phi^{T=1}, p_\Phi^{T=0}) + 2A_y \tag{41}$$

*Proof.* Inequality (38) is by Lemma 5, inequality (39) is by Lemma 1 and Lemma 3; Inequality (40) is by Lemma 6, inequality (41) is by Lemma 1 and Lemma 3; □

**Theorem 2.** *Let $\Phi : \mathcal{X} \to \mathcal{R}$ be an invertible representation with $\Psi$ being its inverse. Let $p_\Phi^{T=1}(\mathbf{r})$, $p_\Phi^{T=0}(\mathbf{r})$ be as defined before. Let $h : \mathcal{R} \times \{0, 1\} \to \mathcal{Y}$, $u := Pr(T = 1)$ and $\mathcal{H}$ be the family of binary functions.*

Let loss function $L$ be the squared loss such that $L(y_1, y_2) = (y_1 - y_2)^2$. Then we have:

$$\epsilon_{PEHE}(h, \Phi)$$

$$\leq 2(\epsilon_{CF}(h, \Phi) + \epsilon_F(h, \Phi) - 2\sigma_y^2) \tag{42}$$

$$\leq 2(\epsilon_F^{T=1}(h, \Phi) + \epsilon_F^{T=0}(h, \Phi) + \frac{K}{2} d_\mathcal{H}(p_\Phi^{T=1}, p_\Phi^{T=0}) - 2\sigma_y^2) \tag{43}$$

Let loss function $L$ be the absolute loss such that $L(y_1, y_2) = |y_1 - y_2|$. Then we have:

$$\epsilon_{PEHE}(h, \Phi)$$

$$\leq \epsilon_{CF}(h, \Phi) + \epsilon_F(h, \Phi) + 2A_y \tag{44}$$

$$\leq \epsilon_F^{T=1}(h, \Phi) + \epsilon_F^{T=0}(h, \Phi) + \frac{K}{2} d_\mathcal{H}(p_\Phi^{T=1}, p_\Phi^{T=0}) + 2A_y \tag{45}$$

*Proof.* Inequality (42) is by Lemma 5, inequality (43) is by Lemma 1 and Lemma 4; Inequality (44) is by Lemma 6, inequality (45) is by Lemma 1 and Lemma 4; □

Obviously, when using Wasserstein distance, there are various versions of bounds for different loss functions as long as they satisfy the (relaxed) triangle inequality and assumptions about $\ell_{h,\Phi}$ in Theorem 1. Similarly, when using $\mathcal{H}$-divergence, there are also various versions of bounds for loss functions that satisfy Assumption 1 and the (relaxed) triangle inequality.

For an empirical sample and a family of representations and hypotheses, we can further upper bound $\epsilon_F^{T=0}$ and $\epsilon_F^{T=1}$ by their respective empirical losses and a model complexity term using standard arguments (Shalev-Shwartz & Ben-David, 2014). Both the Wasserstein distance and $\mathcal{H}$-divergence can be consistently estimated from finite samples (Sriperumbudur et al., 2012; Ben-David et al., 2006; 2010).

### A.4 Illustrative examples

**Examples for the motivation for decomposed patterns.** To explain the dilemma between representation balancing and outcome prediction, we give an intuitive examples and the analytical explanation below to help readers better understand the motivation and importance of involving decomposed patterns in representation balancing models.

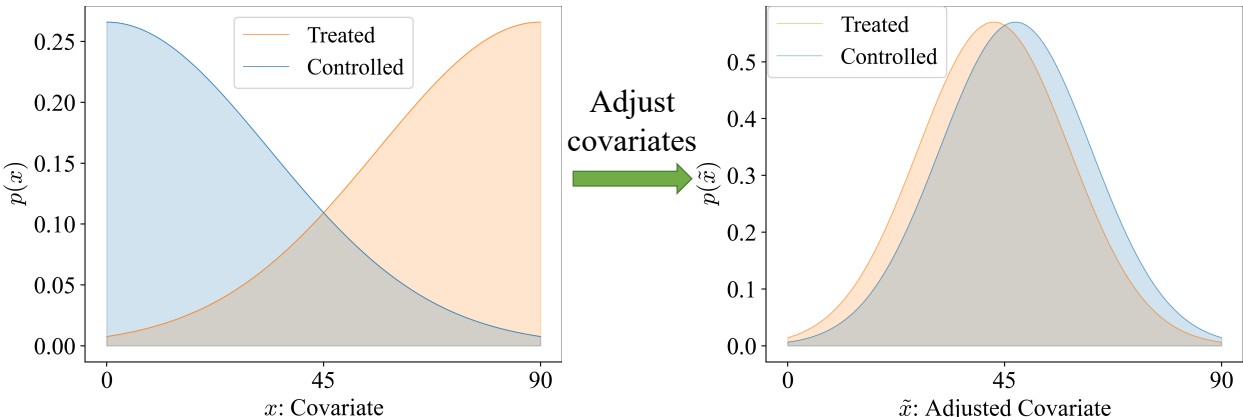

Figure 7: Example for illustrating the importance of decomposed patterns.

**Example 1.** Suppose there is a vaccine to prevent some kind of disease. Let $X$ denote the covariate (age), $T = 1$ denote the treatment (getting vaccinated), $T = 0$ denote the control (not getting vaccinated), and $Y$ denote the outcome (probability of getting the disease). Suppose that the vaccine is assigned according to age, and we have found that the older, the higher the probability of getting the disease. The left graph in Figure 7 shows the distribution of pre-balancing covariate $X$ for treated and controlled groups, which indicates that vaccines are more likely to distribute to older people. Technically, the pre-balancing data preserve the outcome-predictive information: if we want to estimate $Y$ using the covariate $X$, we are confident that people in the treatment/control group (orange/blue) are susceptible/unsusceptible to the disease since they are older/younger. The right panel of Figure 7 shows the distribution of the adjusted covariate $\tilde{X}$, over which the distributions of treated and controlled groups are highly balanced. In this case, however, the distribution of $\tilde{X}$ is too balanced, making it hard to distinguish the treatment samples from the control samples. Consequently, if we want to estimate $Y$ using $\tilde{X}$, we may get confused about which group is susceptible to the disease because the distributions of $\tilde{X}$ are almost identical between the treated and controlled groups. Therefore, only considering balancing patterns can result in a loss of outcome-predictive information.

**Analytical explanation.** Recall that, as defined in Definition 1, $\Phi : \mathcal{X} \to \mathcal{R}$ is a representation function, and $h : \mathcal{R} \times \{0, 1\} \to \mathcal{Y}$ is an outcome function such that $h(\Phi(\mathbf{x}), t)$ estimates $y^t$. For notational simplification, we define $\mathcal{F}$ as a function family consisting $f = h \circ \Phi : \mathcal{X} \times \{0, 1\} \to \mathcal{Y}$ such that $f(\mathbf{x}, t)$ estimates $y^t$. Then, as demonstrated in equation 9, the goal of outcome prediction is to learn the global optimal outcome predictor $f^*$ such that

$$f^* = \operatorname*{argmin}_{f \in \mathcal{F}} \ \mathcal{L}(\mathbf{x}, \mathbf{t}, \mathbf{y}; f).$$

For models without pre-balancing representations such as GNet (Figure 2a) and INet (Figure 2b), the outcome is predicted using only balancing patterns. Specifically, their objectives incorporate the constraint of distance minimization between treatment and control groups over $\Phi$, i.e., $\text{dist}(p_\Phi^{T=1}, p_\Phi^{T=0}) \leq C$ for some small constant $C$, where "dist" can be "Wass" for GNet or $d_\mathcal{H}$ for INet. We define $\mathcal{F}'$ as a function family consisting $f = h \circ \Phi : \mathcal{X} \times \{0, 1\} \to \mathcal{Y}$ such that $f(\mathbf{x}, t)$ estimates $y^t$ with $\Phi$ simultaneously minimizing $\text{dist}(p_\Phi^{T=1}, p_\Phi^{T=0})$. It is obvious that $\mathcal{F}' \subseteq \mathcal{F}$. The outcome predictor for models without pre-balancing representations is learned by

$$f^{**} = \operatorname*{argmin}_{f \in \mathcal{F}'} \ \mathcal{L}(\mathbf{x}, \mathbf{t}, \mathbf{y}; f).$$

Due to $\mathcal{F}' \subseteq \mathcal{F}$, we have $\mathcal{L}(\mathbf{x}, \mathbf{t}, \mathbf{y}; f^*) \leq \mathcal{L}(\mathbf{x}, \mathbf{t}, \mathbf{y}; f^{**})$. That is, if the outcome predictor is only learned by balancing patterns, the learned outcome predictor $f^{**}$ will be suboptimal of $f^*$. Instead, compared to $f^{**}$ learned by only balancing patterns, the outcome predictor learned by both pre-balancing and balancing patterns can give a better estimate, as the corresponding feasible region will be larger than $\mathcal{F}'$.

In summary, on the one hand, involving representation balancing can benefit treatment effect estimation. On the other hand, if $p_\Phi^{T=1}$ and $p_\Phi^{T=0}$ are too balanced, a model may fail to preserve pre-balancing information that is useful to outcome predictions. Such a dilemma motivates us to incorporate PPBR and PDIG such that PPBR improves outcome prediction without harming representation balancing, and PDIG helps a model to achieve more balanced representations without harming outcome prediction.

## A.5 Additional Experimental details

**Additional results on Twins Benchmark.** To investigate the applicability of our model DIGNet to benchmark datasets beyond the commonly used IHDP benchmark, we conducted additional comparisons with several baseline models, including linear, tree, matching, and representation learning methods, on the Twins benchmark, as presented in Table 4.

The Twins dataset comprises records of twin births in the USA between 1989 and 1991. After preprocessing, each unit contains 30 covariates relevant to parents, pregnancy, and birth. The treatment $D = 1$ indicates the heavier twin, while $D = 0$ indicates the lighter twin. The binary outcome variable $Y$ represents 1-year mortality. For more comprehensive details on this dataset and the limitation of IHDP, refer to Curth et al. (2021).

Notably, for $\epsilon_{ATE}$, the simple linear or matching estimator performs best across different methods. On the other hand, when assessing ITE performance using the AUC of potential outcomes, representation learning models all demonstrate strong performance, with AUC values exceeding 0.800 on both training and test sets. Among all the models, our DIGNet achieves the highest AUC results. This observation might stem from the fact that representation balancing models are based on ITE error bounds, rather than ATE error bounds, thereby optimizing for AUC instead of $\epsilon_{ATE}$. This, in turn, inspires us to explore ATE error bounds based on IPM and $\mathcal{H}$-divergence in future research.

Table 4: Training- & test- set AUC & $\epsilon_{ATE}$ on Twins. Mean $\pm$ standard error of 100 runs.

| | Training set | | Test set | |
|---|---|---|---|---|
| | AUC | $\epsilon_{ATE}$ | AUC | $\epsilon_{ATE}$ |
| OLS/LR$_1$ Johansson et al. (2016) | $.660 \pm .005$ | $.004 \pm .003$ | $.500 \pm .028$ | $.007 \pm .006$ |
| OLS/LR$_2$ Johansson et al. (2016) | $.660 \pm .004$ | $.004 \pm .003$ | $.500 \pm .016$ | $.007 \pm .006$ |
| k-NN Crump et al. (2008) | $.609 \pm .010$ | $\mathbf{.003 \pm .002}$ | $.492 \pm .012$ | $\mathbf{.005 \pm .004}$ |
| BART Chipman et al. (2010) | $.506 \pm .014$ | $.121 \pm .024$ | $.500 \pm .011$ | $.127 \pm .024$ |
| CEVAE Louizos et al. (2017) | $.845 \pm .003$ | $.022 \pm .002$ | $.841 \pm .004$ | $.032 \pm .003$ |
| SITE Yao et al. (2018) | $.862 \pm .002$ | $.016 \pm .001$ | $.853 \pm .006$ | $.020 \pm .002$ |
| BLR Johansson et al. (2016) | $.611 \pm .009$ | $.006 \pm .004$ | $.510 \pm .018$ | $.033 \pm .009$ |
| BNN Johansson et al. (2016) | $.690 \pm .008$ | $.006 \pm .003$ | $.676 \pm .008$ | $.020 \pm .007$ |
| TARNet Shalit et al. (2017) | $.849 \pm .002$ | $.011 \pm .002$ | $.840 \pm .006$ | $.015 \pm .002$ |
| CFR-Wass (GNet) Shalit et al. (2017) | $.850 \pm .002$ | $.011 \pm .002$ | $.842 \pm .005$ | $.028 \pm .003$ |
| DIGNet (Ours) | $\mathbf{.874 \pm .001}$ | $.004 \pm .001$ | $\mathbf{.871 \pm .001}$ | $.008 \pm .001$ |

**Hyperparameters.** In simulation studies, we ensure a fair comparison by fixing all the hyperparameters in all datasets across different models. The relevant details are stated in Table 5. In IHDP studies, to compare

Table 5: Hyperparameters of different models in simulation studies.

| | $\Phi_E$ | $\Phi_G$ | $\Phi_I$ | $\pi$ | $h^1$ | $h^0$ | $\alpha_1$ | $\alpha_2$ | batchsize | iteration | learning rate | learning rate for $\pi$ |
|---|---|---|---|---|---|---|---|---|---|---|---|---|
| Gnet | (100, 100, 100, 100) | – | – | – | (100, 100) | (100, 100) | 0.1 | – | 100 | 300 | $1e^{-3}$ | – |
| Inet | (100, 100, 100, 100) | – | – | (100, 100, 100) | (100, 100) | (100, 100) | – | 0.1 | 100 | 300 | $1e^{-3}$ | $1e^{-4}$ |
| DGNet | (100, 100, 100, 100) | (100, 100) | – | – | (100, 100) | (100, 100) | 0.1 | – | 100 | 300 | $1e^{-3}$ | – |
| DINet | (100, 100, 100, 100) | – | (100, 100) | (100, 100, 100) | (100, 100) | (100, 100) | – | 0.1 | 100 | 300 | $1e^{-3}$ | $1e^{-4}$ |
| DIGNet | (100, 100, 100, 100) | (100, 100) | (100, 100) | (100, 100, 100) | (100, 100) | (100, 100) | 0.1 | 0.1 | 100 | 300 | $1e^{-3}$ | $1e^{-4}$ |

with the baseline model CFR-Wass (GNet), we remain the hyperparameters of INet, DGNet, DINet and the early stopping rule the same as those used in CFR-Wass Shalit et al. (2017). Since DIGNet is more complex than other four models, we adjust the hyperparameters of $\Phi_E$, $\Phi_G$, $\Phi_I$, $\alpha_1$, and $\alpha_2$ for DIGNet as Shalit et al. (2017) do. The relevant details are stated in Table 6.

Table 6: Hyperparameters of different models in IHDP experiments.

| | $\Phi_E$ | $\Phi_G$ | $\Phi_I$ | $\pi$ | $h^1$ | $h^0$ | $\alpha_1$ | $\alpha_2$ | batchsize | iteration | learning rate | learning rate for $\pi$ |
|---|---|---|---|---|---|---|---|---|---|---|---|---|
| Gnet | $(100,100,100,100)$ | – | – | – | $(100,100,100)$ | $(100,100,100)$ | 1 | – | 100 | 600 | $1e^{-3}$ | – |
| Inet | $(100,100,100,100)$ | – | – | $(200,200,200)$ | $(100,100,100)$ | $(100,100,100)$ | – | 1 | 100 | 600 | $1e^{-3}$ | $1e^{-3}$ |
| DGNet | $(100,100,100,100)$ | $(100,100)$ | – | – | $(100,100,100)$ | $(100,100,100)$ | 1 | – | 100 | 600 | $1e^{-3}$ | – |
| DINet | $(100,100,100,100)$ | – | $(100,100)$ | $(200,200,200)$ | $(100,100,100)$ | $(100,100,100)$ | – | 1 | 100 | 600 | $1e^{-3}$ | $1e^{-3}$ |
| DIGNet | $(100,100,100,100,100,100)$ | $(100,100,100)$ | $(100,100,100)$ | $(200,200,200)$ | $(100,100,100)$ | $(100,100,100)$ | 0.1 | 1 | 100 | 600 | $1e^{-3}$ | $1e^{-3}$ |

**Analysis of training time and training stability.** We record the time it took for different models to run through 100 IHDP datasets, and each model is trained within 600 epochs. Following Shalit et al. (2017), all models adopt the early stopping rule. We also record the average early stopping epoch on 100 runs and the actual time on 100 runs, where (actual time) = (total time) × (average early stopping epoch)/600. Not surprisingly, GNet took the least amount of time with 3096 seconds since the objective of GNet is the simplest. However, it is very interesting that the proposed methods, DGNet and DINet, are the first two to early stop. As a result, though DGNet and DINet have multi-objectives, they spent less actual training time but achieved better ITE estimation compared to GNet and INet. Since GNet and INet are actually DGNet and DINet with PPBR ablated, we find that PPBR component can help a model achieve better ITE estimates with less time. In addition, we find that DIGNet spent the longest time to optimize since it has the most complex objective. To further study the stability of the model training, we also plot the metrics $\sqrt{\epsilon_F}$, Wass, $\hat{d}_{\mathcal{H}}$, and $\sqrt{\epsilon_{PEHE}}$ for the first 100 epochs of each model on the first IHDP dataset. We find that the training process of DIGNet is stable, even steadier than GNet and INet. From this perspective, we haven't seen a difficulty of optimizing DIGNet.

Table 7: Training time records on 100 IHDP datasets.

| Model | Time for 600 epochs | Avg early stopping | Actual time | $\sqrt{\epsilon_{PEHE}}$ on test set |
|---|---|---|---|---|
| GNet | 3096s | 240.61 | 1241s | 0.77±0.18 |
| INet | 4042s | 254.19 | 1712 | 0.72±0.11 |
| DGNet | 3775s | 169.17 | 1064s | 0.60±0.09 |
| DINet | 3212s | 157.98 | 846s | 0.60±0.11 |
| DIGNet | 4984s | 226.76 | 1884s | 0.45±0.04 |

## A.6 Objectives of Different Models

**Objective of GNet.**

$$\min_{\Phi_E, h^t} \quad \mathcal{L}_y(\mathbf{x}, \mathbf{t}, \mathbf{y}; \Phi_E, h^t) + \alpha_1 \mathcal{L}_G(\mathbf{x}, \mathbf{t}; \Phi_E).$$

**Objective of INet.**

$$\max_{\pi} \quad \alpha_2 \mathcal{L}_I(\mathbf{x}, \mathbf{t}; \Phi_E, \pi),$$
$$\min_{\Phi_E, h^t} \quad \mathcal{L}_y(\mathbf{x}, \mathbf{t}, \mathbf{y}; \Phi_E, h^t) + \alpha_2 \mathcal{L}_I(\mathbf{x}, \mathbf{t}; \Phi_E, \pi).$$

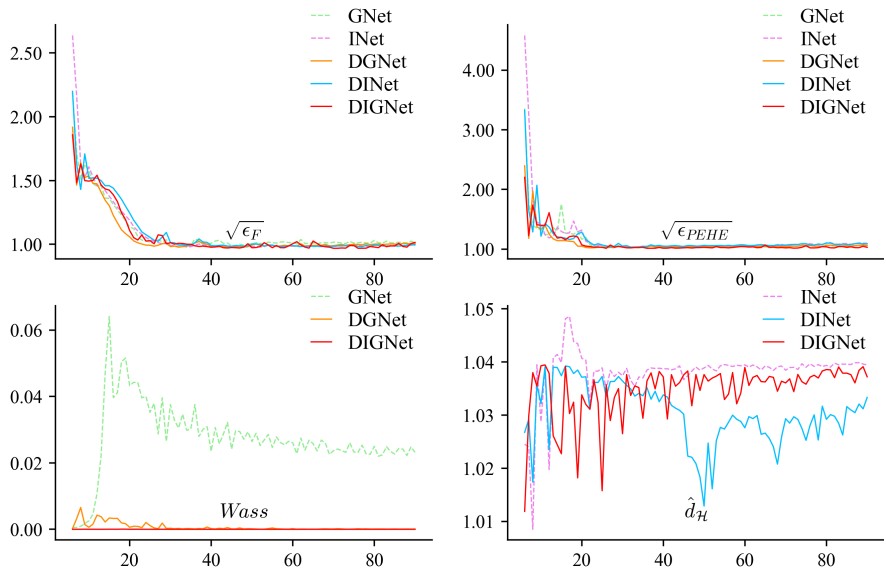

Figure 8: Training loss plots for the first 100 epochs on the first IHDP dataset.

**Objective of DINet.**  Note that similar to DIGNet, the pre-balancing patterns are preserved by only updating $\Phi_I$ but fixing $\Phi_E$ in the second step.

$$\max_{\pi} \quad \alpha_2 \mathcal{L}_I(\mathbf{x}, \mathbf{t}; \Phi_I \circ \Phi_E, \pi),$$

$$\min_{\Phi_I} \quad \alpha_2 \mathcal{L}_I(\mathbf{x}, \mathbf{t}; \Phi_I \circ \Phi_E, \pi),$$

$$\min_{\Phi_E, \Phi_I, h^t} \quad \mathcal{L}_y(\mathbf{x}, \mathbf{t}, \mathbf{y}; \Phi_E \oplus (\Phi_I \circ \Phi_E), h^t).$$

**Objective of DGNet.**  Note that similar to DIGNet, the pre-balancing patterns are preserved by only updating $\Phi_G$ but fixing $\Phi_E$ in the first step.

$$\min_{\Phi_G} \quad \alpha_1 \mathcal{L}_G(\mathbf{x}, \mathbf{t}; \Phi_G \circ \Phi_E),$$

$$\min_{\Phi_E, \Phi_G, h^t} \quad \mathcal{L}_y(\mathbf{x}, \mathbf{t}, \mathbf{y}; \Phi_E \oplus (\Phi_G \circ \Phi_E), h^t).$$

**Objective of DIGNet.**

$$\min_{\Phi_G} \quad \alpha_1 \mathcal{L}_G(\mathbf{x}, \mathbf{t}; \Phi_G \circ \Phi_E),$$

$$\max_{\pi} \quad \alpha_2 \mathcal{L}_I(\mathbf{x}, \mathbf{t}; \Phi_I \circ \Phi_E, \pi),$$

$$\min_{\Phi_I} \quad \alpha_2 \mathcal{L}_I(\mathbf{x}, \mathbf{t}; \Phi_I \circ \Phi_E, \pi),$$

$$\min_{\Phi_E, \Phi_I, \Phi_G, h^t} \quad \mathcal{L}_y(\mathbf{x}, \mathbf{t}, \mathbf{y}; \Phi_E \oplus (\Phi_I \circ \Phi_E) \oplus (\Phi_G \circ \Phi_E), h^t).$$

