# OpenReview forum: "Representation Balancing with Decomposed Patterns for Treatment Effect Estimation"
_TMLR — Rejected by TMLR_

### Review · Reviewer_ZjWZ · 2023-03-25

**Summary Of Contributions:**

The paper proposes a new representation learning method for estimating treatment effects from observational data. The representations are optimized to be (1) balanced between the treated and control populations and (2) predictive of the factual outcomes. The first objective is achieved by adding a regularizer to encourage matching between the two populations, i.e., which is called Patterns into Decompositions of Individual propensity confusion and Group distance minimization (PDIG). The second objective is attained by also giving the pre-balanced representation to the network for making predictions, which is called decomposing proxy features into Patterns of Pre-balancing and Balancing Representations (PPBR). Theoretical analysis is provided for the (PDIG). Empirical evidence is provided to show the performance gain when both techniques are combined together.

**Audience:**

Yes

**Broader Impact Concerns:**

N.A.

**Claims And Evidence:**

Yes

**Requested Changes:**

1. The most important adjustment needed is to provide an analysis when multiple techniques (especially the ones in Section 4) proposed in the paper are combined. Perhaps the trade-off problem does not have a general solution. The authors should propose new distributional assumptions under which their method works better than the others.
2. Some important terminology in the paper is incorrect. For example, what the author called "Individual treatment effect" in the paper is often called "Individiualized treatment effect" or more often "conditional average treatment effect" in the literature. "Individual treatment effect" is often used to denote the random variable Y_i(1)-Y(0).
3. Improve the writing for the experiment part. I suggest only reporting the performance of treatment effect estimation in the main paper while reporting the others in the Appendix.

**Strengths And Weaknesses:**

Strengths:

1. Most part of the paper is well-organized and written.
2. The paper studies a well-known and important problem in the topic, which is how to deal with the trade-off between balancing and predictive power.
3. The experiment section is very intensive and informative.


Weakness:

1. The theoretical result (Section 3.1) is incremental on top of the results in Shalit et al., 2017 and Johansson et al., 2022.
2. Using propensity score for matching population (Section 3.2) is not a new idea even via deep neural networks; see the reference [A] below and the references therein. The theory in the section is mainly from the domain adaption literature.
3. Even though theoretical analysis is given for the techniques in Section 3. No analysis is given to the combination of them. This is important because it is the key to solving the trade-off problem.
4. The experimental section is hard to read.


[A] Kallus, N. (2020). Deepmatch: Balancing deep covariate representations for causal inference
using adversarial training.  In International Conference on Machine Learning

---

> ### Author Response · Authors · 2023-06-27
> **Response to Reviewer ZjWZ (1/3)**
>
> Dear Reviewer ZjWZ,
>
> Thank you for your valuable feedback on our paper. We sincerely appreciate your time and effort in reviewing our work, as well as your positive comments regarding our writing, presentation, and comprehensive experimental results. We have carefully considered your comments and made the necessary revisions and responses accordingly. The revised manuscript has been uploaded and the changes are colored in ```RED```. We hope that these changes and our responses can adequately address your concerns.
>
> ---
> **Concern 1& Concern 2:**
>
> We greatly appreciate your comments and suggestions regarding our theoretical contributions.
>
> Regarding Section 3.1, we acknowledge that the theoretical results presented in our paper are built upon the foundations laid out in Shalit et al. (2017) [1]. However, our extension considers a more general case where the loss function can accommodate any symmetric loss. We agree with the reviewer's observation that this theoretical result is somewhat incremental in this research field.
>
> Regarding Section 3.2, we would like to clarify that in our paper, the propensity score is not used for matching populations. As mentioned in the first paragraph of this section, the propensity score is used to identify if representations are adequately balanced. Thus our proposed concept of "propensity confusion" provides a natural interpretation and rationale for minimizing the $\mathcal{H}$-divergence-based ITE error bound Eqn. (6). We have revised the corresponding content in Section 3.2 to make this point clearer. We apologize if our original expressions caused any confusion regarding the role of the propensity score.
>
> Furthermore, we would like to express our gratitude to the reviewer for highlighting the connection with the domain adaptation literature. But please allow us to emphasize that our theoretical results are not a direct or trivial extension of the domain adaptation literature. As stated in [1], the theoretical foundations of causal representation balancing works, including the papers cited in Section 1.1, are closely connected to the domain adaptation literature (e.g., [2]). Nevertheless, the core of the causal representation balancing theory and the corresponding proof details are distinct from the domain adaptation literature for the following reasons:
>
> 1. The quantity of interest is the ITE error, not a classification error. So the target variable can be either continuous or discrete.
> 2. The main challenge lies in counterfactual estimation of all units in the same dataset, rather than dealing with unlabeled units in another target dataset.
> 3. Most importantly, the proof techniques and details of the theoretical upper bound of the ITE error using $\mathcal{H}$-divergence (Lemma 4 in the Appendix) significantly differs from the theoretical results presented in [2].
>
> In addition, we do follow the strategy proposed in [3] to empirically approximate the $\mathcal{H}$-divergence, and we have emphasized this point in the revised paper. Thank you for your comments that allow us to make our theoretical contributions clearer!

---

> > ### Author Response · Authors · 2023-06-27
> > **Response to Reviewer ZjWZ (2/3)**
> >
> > **Concern 3 & Request Change 1:**
> >
> > Thank you for pointing out the importance of theoretical analysis, and we strongly agree that analyzing the combination of the proposed techniques theoretically will be crucial for solving the trade-off problem. However, as mentioned in Section 6, the theoretical solution to the trade-off between domain invariance and domain discrimination remains a common open challenge in both domain adaptation and causal representation balancing literature. We believe it would be interesting to consider developing new distributional divergences/assumptions (as you suggested) and finding a new upper bound that is robust to the trade-off issue (possibly in the combination of distributionally robust optimization and nonparametric inference methods). We have updated Section 6 to highlight these two exciting directions for future research.
> >
> > Given the lack of a well-established theoretical solution in the current research field, our paper focuses on providing effective empirical solutions to the trade-off challenge. We hope that these empirical insights can shed light on future theoretical studies. Our approach is motivated by the rationale behind causal representation balancing: achieving a smaller ITE error relies on (1) accurate estimation of factual outcomes and (2) minimizing the distributional discrepancy between treated and controlled groups. To address these objectives, we investigate the components PDIG and PPBR, which respectively enhance treatment effect estimation by (i) learning more balancing patterns (task 2) without affecting factual outcome prediction (task 1), and (ii) improving factual outcome prediction (task 1) without affecting learning balancing patterns (task 2), respectively.
> >
> > To verify the effectiveness of the proposed PDIG and PPBR, we conduct extensive empirical analysis through ablation studies on both simulated and benchmark datasets, as presented in Section 5.2. In the next response, we will further demonstrate the necessity and significance of our experimental results, which serve as important empirical analyses of our proposed methods.
> >
> > ---
> >
> > **Concern 4 & Request Change 3**
> >
> > We appreciate your valuable feedback regarding the presentation of our experimental results. We have carefully considered your suggestion of including only the treatment effect estimation (ITE) results in the main paper. However, we believe that the current content in the main paper provides a comprehensive analysis of PDIG and PPBR, which helps us gain insights into how these techniques enhance the estimation process through the two aspects mentioned earlier. The detailed analysis can be found in Section 5.2, where we examine the source of gain.
> >
> > For instance, in the ablation study for PDIG, we compare DIGNet with DGNet and DINet (Figure 4) and observe that DIGNet generally exhibits similar factual error $\epsilon_F$ to its ablated models DGNet and DINet. However, it consistently achieves a smaller $\mathcal{H}$-divergence (or Wasserstein distance) compared to DINet (or DGNet). As a result, DIGNet ultimately achieves a smaller counterfactual error than DGNet and DINet. This analysis indicates that PDIG enhances ITE estimation by learning more balancing patterns (task 2) without affecting factual outcome prediction (task 1). A similar analysis has also been conducted for PPBR in this section, where PPBR is verified to enhance ITE estimation by improving factual outcome prediction (task 1) without affecting the learning of balancing patterns (task 2).
> >
> > Based on these findings, we believe that retaining the current analysis of all the figures and tables would provide comprehensive information to verify the effectiveness of PDIG and PPBR. We also re-organize unclear expressions in the experimental section. We hope that our explanation and revision would clarify the reason for including the relevant analysis in the main paper.

---

> > > ### Author Response · Authors · 2023-06-27
> > > **Response to Reviewer ZjWZ (3/3)**
> > >
> > > **Request Change 2**
> > >
> > > We strongly agree the reviewer that $Y_i(1)-Y_i(0)$ is the most appropriate definition of individual treatment effect for some individual $i$, and some machine learning works [10-12] also use this notation. The quantity we call ITE for individual $i$ is defined as $\mathbb{E}[ Y_i(1)-Y_i(0)|\mathbf{X}]$, which is usually called Conditional Average Treatment Effect (CATE) in nonparametric statistics literature, and some machine learning works [8, 9] also call this term as CATE.
> > >
> > > We call the quantity $\mathbb{E}[ Y_i(1)-Y_i(0)|\mathbf{X}]$ as ITE just in order to keep the consistency with the line of causal representation balancing literature (e.g., [1, 4-7]). In case of any confusion, we have emphasized that $Y_i(1)-Y_i(0)$ is the original definition of ITE, but following the definition in [1], we denote ITE by the CATE formulation $\mathbb{E}[ Y_i(1)-Y_i(0)|\mathbf{X}]$. Thank you for your thoughtful suggestion!
> > >
> > >
> > > Finally, we would like to thank the reviewer again for spending your valuable time reviewing our paper! Your comments are constructive for our novelty, contribution, and promising future research. We hope that our responses and the revised paper have sufficiently addressed any concerns regarding the novelty. We are open to further discussions and welcome any additional comments or suggestions you may have!
> > >
> > > ---
> > >
> > > [1] Uri Shalit, Fredrik D Johansson, and David Sontag. Estimating individual treatment effect: generalization bounds and algorithms. In International Conference on Machine Learning, pp. 3076–3085. PMLR, 2017.
> > >
> > > [2] Shai Ben-David, John Blitzer, Koby Crammer, Alex Kulesza, Fernando Pereira, and Jennifer Wortman Vaughan. A theory of learning from different domains. Machine learning, 79(1):151–175, 2010.
> > >
> > > [3] Yaroslav Ganin, Evgeniya Ustinova, Hana Ajakan, Pascal Germain, Hugo Larochelle, François Laviolette, Mario Marchand, and Victor Lempitsky. Domain-adversarial training of neural networks. The journal of machine learning research, 17(1):2096–2030, 2016.
> > >
> > > [4] Fredrik D. Johansson, Uri Shalit, Nathan Kallus, and David Sontag. Generalization bounds and representation learning for estimation of potential outcomes and causal effects. Journal of Machine Learning Research, 23(166):1–50, 2022.
> > >
> > > [5] Serge Assaad, Shuxi Zeng, Chenyang Tao, Shounak Datta, Nikhil Mehta, Ricardo Henao, Fan Li, and
> > > Lawrence Carin. Counterfactual representation learning with balancing weights. In International Conference on Artificial Intelligence and Statistics, pp. 1972–1980. PMLR, 2021.
> > >
> > > [6] Ruocheng Guo, Jundong Li, and Huan Liu. Learning individual causal effects from networked observational data. In Proceedings of the 13th International Conference on Web Search and Data Mining, pp. 232–240, 2020c.
> > >
> > > [7] Liuyi Yao, Sheng Li, Yaliang Li, Mengdi Huai, Jing Gao, and Aidong Zhang. Representation learning for treatment effect estimation from observational data. Advances in Neural Information Processing Systems, 31, 2018.
> > >
> > > [8] Xin Du, Lei Sun, Wouter Duivesteijn, Alexander Nikolaev, and Mykola Pechenizkiy. Adversarial balancing-based representation learning for causal effect inference with observational data. Data Mining and Knowledge Discovery, 35(4):1713–1738, 2021.
> > >
> > > [9] Alicia Curth and Mihaela van der Schaar. On inductive biases for heterogeneous treatment effect estimation. Advances in Neural Information Processing Systems, 34:15883–15894, 2021.

---

### Review · Reviewer_KtuW · 2023-04-27

**Summary Of Contributions:**

The authors solve for the standard conditional average treatment effects problem: predict the expected difference of potential outcomes of a treatment, given a set of covariates. In particular, the authors introduce a family of methods based on the observation that the treatment effects problem can be decomposed in two subtasks: deconfound the covariates to infer unbiased effect estimates, and accurately predict outcomes.

In the paper the authors propose some new bounds on the PEHE (based on previous work by Shalit et al.) and use insights from these to motivate their proposal: DIGNet (and variants). DIGNet is validated on the well adopted IHDP dataset and found to be outperforming a wide range of existing proposals.

**Audience:**

Yes

**Claims And Evidence:**

No

**Requested Changes:**

Some suggestions which I think would improve the readability of this paper. Note that I find readibility to be an issue and would emphasise that these requests are quite important for this particular paper.

- please name your definitions. Not doing so makes the paper really hard to read and does not easily relate to the surrounding text.

- it seems that most of the definitions( I.g. H-divergence or IPM) are quite standard and would suggest they are moved to an appendix to improve readability.

- the method builds on theorem 2, I believe it would be nicer to combine sec 3 and 4. The authors first present some theory which has no relevance to the paper until sec 4 which makes the paper hard to follow. I believe it would be nicer to put the theorems in proper context (the methods section in this case).

- Some illustrative figures would greatly improve readability of the paper.

**Strengths And Weaknesses:**

STRENGTHS

- the paper is well-situated in an interesting area of research. I don't think it needs further emphasising that treatment effects are an important area with large potential for adoption in practice

- I find the authors' observation to be interesting and sensible. It seems very logical to me to base a method around the two tasks the authors define (as indicated in the previous section in my review)

- The literature review seems extensive and the benchmarks are very representative of contemporary literature


WEAKNESSES

- IHDP has been heavily criticised. Does the proposed method also work for other data? (See Curth et al. (NeurIPS D&B, 2021)). I have indicated below that the claims are currently not backed up by evidence. For context, this comment is the reason for selecting "no" on that yes/no question.

- Fig. 3 & 4 could use some context. How do other methods- not proposed in this paper -compare (beyond GNet which could be argued as a special case of the framework proposed by the authors)?


REFS

Curth et al. https://openreview.net/forum?id=FQLzQqGEAH

---

> ### Author Response · Authors · 2023-07-25
> **Response to Reviewer KtuW (1/2)**
>
> Dear Reviewer KtuW,
>
>
> We are thankful for your valuable feedback and the time you've taken to review our paper. We are delighted and grateful for your recognition of the significance, idea, literature, and benchmarks of our studies, which greatly encourages us for further revision and future research. After carefully considering your comments, we have made corresponding changes in the revised manuscript. The updated version has been uploaded, and all the modifications are indicated in ```RED```. Below, we provide a point-by-point response to your concerns and requested changes.
>
> * Weaknesses
>
> **1.**  We acknowledge the reviewer's comment regarding the IHDP benchmark dataset. Although IHDP is the most widely used benchmark in causal representation learning works, we agree with the reviewer that IHDP may have a limitation due to its semi-synthetic outcome generation process. In response to this concern, we consider conducting additional Twins experiments as supplementary results, because Twins dataset contains "all potential outcomes" data, as stated in Section 4.2 of the reference provided by the reviewer. We believe that incorporating the Twins dataset in our comparisons would mitigate the possibility of specific data generating processes favoring certain methods and better validate the effectiveness of our model. Thus, in addition to the commonly used IHDP benchmark, we performed additional comparisons with several baseline models, including linear, tree, matching, and representation learning methods, on Twins datasets. These results are presented in Table 4 of Section A.5.
>
> We briefly introduced the Twins dataset and cited the reference (Curth et al., 2021) to introduce Twins as a supplementary dataset and to highlight the potential limitations of relying solely on IHDP. The findings from the Twins dataset indicate that for $\epsilon_{ATE}$, the simple linear or matching estimator performs best across different methods. On the other hand, when assessing ITE performance using the AUC of potential outcomes, representation learning models all demonstrate strong performance, with AUC values exceeding $0.800$ on both training and test sets. Among all the models, our DIGNet achieves the highest AUC results. This observation might stem from the fact that representation balancing models are based on ITE error bounds, rather than ATE error bounds, thereby optimizing for AUC instead of $\epsilon_{ATE}$. This, in turn, inspires us to explore ATE error bounds based on IPM and $\mathcal{H}$-divergence in future research.
>
>
> **2.**  Thank you for your thoughtful suggestion! First, please allow us to emphasize that Figures 3 and 4 (Figures 4 and 5 in the revised version) primarily serve as ablation studies to investigate the effectiveness of the important contributions of our proposed PPBR and PDIG components. Specifically, by comparing DIGNet with DGNet (or DIGNet with DINet), we can assess the effectiveness of PDIG; by comparing DGNet with GNet (or DINet with INet), we can test the effectiveness of PPBR.
>
> For instance, the Figures reveal that DGNet and GNet (or DINet and INet) exhibit comparable performance in terms of Wasserstein distance (or $\mathcal{H}$-divergence), while DGNet (or DINet) achieves smaller factual outcome prediction errors compared to GNet (or INet), with a $|1.07/1.12-1|=4.5$\% (or $|1.07/1.08-1|=0.9$\%) reduction in $\sqrt{\epsilon_{F}}$. This suggests that PPBR can effectively contribute to reducing outcome prediction errors without affecting representation balancing, which results in smaller errors of the ITE upper bound and therefore enhances ITE estimation.
>
> We have carefully considered your suggestion to include other baseline models in Figure 4 and we totally understand that the suggestion aims to enrich our experimental results. Nevertheless, given that the main objective of Figures 4 and 5 is to conduct ablation studies to investigate the impact of PPBR and PDIG on ITE estimation and how they assist in this regard, we believe it would be more appropriate to maintain the current version. This allows us to maintain the focus on understanding the effectiveness of our proposed components in the context of ITE estimation. Please accept our gratitude for your kind suggestion!

---

> ### Author Response · Authors · 2023-07-25
> **Response to Reviewer KtuW (2/2)**
>
> * Requested Changes
>
> **1.** We sincerely appreciate your suggestion regarding the clarity of definitions. To address this concern, we have revised several parts to enhance the understanding of key quantities and concepts presented in our paper. First, we have expanded Section 2, particularly in the paragraph titled "Notations." Second, we have introduced Figure 1, which visually explains pre-balancing patterns, balancing patterns, PPBR, and PDIG. This visual aid aims to improve the readability and comprehension of these crucial concepts. Third, to provide a clearer understanding of the proposed concept of propensity confusion, we have included more extensive explanations in the third paragraph of the Introduction.
> By incorporating these improvements, we believe that the current version of our paper now provides more clear definitions, thereby enhancing the overall clarity of our work. Thank you for your feedback and valuable input on our manuscript.
>
> **2.** We are thankful for your suggestion regarding moving some of the definitions, such as $\mathcal{H}$-divergence and IPM, to Appendix. However, after careful consideration, we have decided to keep these definitions in the main body of the paper. We think it would be good to include these definitions in the main text as they provide essential context and clarity for readers, which is helpful for them to understand our overall framework quickly. In addition, even though the concepts are fundamental to the field researcher,  we think readers in other areas may not be necessarily familiar with the concepts of causal inference or distributional measures. Presenting these definitions in the main text may make the paper more easily accessible to a broader audience. Further, including these definitions in the main text allows us to better connect the theoretical foundations with the methodology. Thank you again for your thoughtful suggestion!
>
>
> **3.** Thank you for your suggestion regarding the organization of Section 3 and Section 4. As our method is built upon representation balancing with Wasserstein distance and $\mathcal{H}$-divergence, we present the theoretical results based on these measures in Theorem 1 and Theorem 2 in Section 3. Subsequently, we introduce GNet and INet, which are fundamental model structures associated with Wasserstein distance and $\mathcal{H}$-divergence, respectively. We further explain how our proposed PPBR and PDIG components are incorporated into the counterfactual regression neural net to further build DIGNet upon GNet and INet.
>
> Therefore, keeping the current organization of the theoretical section (Section 3) and the method section (Section 4) may provide clarity for readers to understand the empirical contributions and their theoretical foundations separately. However, considering any potential unclear points, we have rewritten Section 3 and part of Section 4 to improve the readability of our paper, with changes highlighted in red. We apologize for any confusion in our original version, and we hope the updated organization will enhance the overall understanding and context of our work.
>
> **4.** Thank you for your valuable suggestions! We have made enhancements to the readability of our paper by introducing Figure 1, which provides a visual explanation of pre-balancing patterns, balancing patterns, PPBR, and PDIG. This visual aid is intended to facilitate better comprehension of these essential concepts. Furthermore, Figure 2 includes illustrations of our model DIGNet and its variants, further helping readers better understand the structure of our proposed approach. We believe that including these illustrative figures based on your suggestions would be very helpful for the overall readability of our paper.
>
> Finally, we would like to thank the reviewer again for spending time reviewing our paper. We have revised our paper according to your valuable comments. We sincerely hope that our responses have well addressed your concerns. If you have any other questions, please feel free to leave comments!

---

### Review · Reviewer_LMUR · 2023-07-11

**Summary Of Contributions:**

This paper addresses the estimation of individual treatment effects (ITE) as well as average treatment effects (ATE) from observational data, where there is covariate shift between the treated and control groups. In terms of theoretical contributions, the paper generalizes existing bounds on counterfactual error and Precision in Estimation of Heterogeneous Effects (PEHE), due mainly to Shalit et al. (2017). These generalizations are from 1-Wasserstein distance to $\mathcal{H}$-divergence and also from squared error to absolute error. In terms of methodology, the paper proposes several neural network architectures for ITE estimation: balancing based on individual propensity confusion (INet), concatenation of balanced representations with pre-balancing representations (DINet, DGNet), and a combination of balancing based on propensity confusion and group distance as well as pre-balancing representations (DIGNet). Experiments are conducted on synthetic data and the semi-synthetic IHDP dataset to evaluate the estimation performance of the various enhanced architectures as well as their sources of improvement. Overall, the results appear to show a significant gain due to including pre-balancing representations and a smaller gain due to combining propensity confusion and group distance balancing.

**Audience:**

Yes

**Claims And Evidence:**

No

**Requested Changes:**

The comments above under Correctness, Experiments, and comments 1-3 under "Presentation of Theoretical Results" are all critical for me.

Among the Clarity issues, the first three are more important.

Among the Minor Comments and Questions, please answer 7 and 4 in particular.

**Strengths And Weaknesses:**

## Strengths
- Balancing by minimizing $\mathcal{H}$-divergence (INet), while a natural generalization, has not been proposed in the treatment effect estimation literature to my knowledge.
- Several other neural architectures are proposed, embodying the ideas of concatenating pre-balancing representations (DINet, DGNet) and combined balancing based on $\mathcal{H}$-divergence and Wasserstein distance minimization (DIGNet).
- The gain due to including pre-balancing representations appears to be significant across both synthetic datasets and the IHDP dataset.

## Weaknesses
- Correctness
    1. A possible issue in the proof of Lemma 4, upon which Theorem 2 depends: The step from (24) to (25) implies that $\ell_{h,\Phi}(\Psi(r), t) \leq K$ for all $r, t$. According to Definition 6, this would require $\int_{\mathcal{Y}} L(y^t, y_2) p(y^t | x) dy^t \leq K$ for all $y_2 = h(r, t)$. This differs however from Assumption 1: $\int_{\mathcal{Y}} L(y_1, y_2) dy_1 \leq K$. Please address this discrepancy.
    1. Eq. (6): I believe the entire right-hand side needs to be multiplied by 2 to be the same as (41) in the appendix.
    1. The proof of Lemma 6 implies that $A_y$ in Definition 12 should be defined as the max, not the min.
- Presentation of Theoretical Results and Relationship to Prior Work
    1. Both Lemma 1 and Theorem 1 are stated with $L$ as the squared loss. Thus I think they should be credited to Shalit et al. (2017). Is Lemma 5 in Appendix A.3 also from Shalit et al.?
    1. It is confusing to have another Theorem 1 in the appendix that differs from the Theorem 1 in Section 3.1. I suggest stating the full theorem from the appendix in Section 3.1, which would also bring the novel element of absolute loss from the appendix into the main paper. Otherwise, the theorem in the appendix should be given a different number to distinguish it. Same comments about Theorem 2.
    1. I went through the proof of Theorem 2 and found it to be similar to the proof of Theorem 1 except for using $\mathcal{H}$-divergence instead of Wasserstein distance to bound the difference between the treated and control distributions. The two theorems also look the same except for $\mathcal{H}$-divergence versus Wasserstein. Thus I think the claim that “our theoretical derivations are very different from … causal representation balancing works (Shalit et al., 2017, etc.)” is overstated.
    1. The first paragraph of Section 3 is also not clear enough that the idea of balancing based on propensity confusion is a contribution of this work, in contrast to previous group distance/IPM-based approaches. If this is accurate, I think this contribution could also be mentioned in the introduction.
- Experiments
    1. Section 5.1, Models and Metrics paragraph: It is stated here that all models share the same values for $(\alpha_1, \alpha_2)$, which are the trade-off parameters between group distance/propensity confusion and outcome prediction. However, couldn’t the different variants (DGNet, DINet, DIGNet, etc.) be optimized at different values of $\alpha_1, \alpha_2$? Is it really a fair comparison to fix the same values for all of them?
    1. Section 5.2, Source of Gain results: I wonder about the statistical significance of the reported improvements, especially DIGNet over DINet. I am not saying that all improvements have to be statistically significant, but rather that it would be good to understand if perhaps the larger improvements are and the smaller ones are not. If the DIGNet-DINet improvement is not statistically significant, then I would also omit “overwhelmingly” from the last sentence of Section 5.
- Clarity
    1. I found all the acronyms (PDIG, PPBR, DIGNet, etc.) confusing as well as their relationships to each other, especially before Section 4.2 and Figure 1. I think a better explanation is needed in the introduction, perhaps using a diagram.
    1. I did not understand what is meant by “patterns” as in “balancing patterns,” “decomposed patterns.” The paper never defines the term.
    1. Propensity confusion is not defined until the first paragraph of Section 3.2. I think this one-sentence definition should be moved or copied to the introduction. It is also not explained how $\mathcal{H}$-divergence is connected to propensity confusion.
    1. Page 2, paragraph 3
        - Line 5: What does “concept” mean here?
        - Lines 6-7: What about “rationality”?
        - Also line 6: The acronym ITE has not been defined yet.
    1. Section 4.1, after (7): The phrase “if a model does not have decomposition modes like GNet and INet” is unclear about whether GNet and INet have or lack decomposition modes, although I believe it is the latter.
    1. Section 4.2, first two lines: How is INet “meaningful and interpretable”?
    1. Table 3: What is meant by “result is not reproducible”?
    1.  Appendix A.1, Lemma 2: “Half of $\mathcal{H}$-divergence” is understandable and correct but I think the lemma should be stated more formally and completely.
- Minor Comments and Questions
    1. Section 2, first line: I think “random variables” is sufficient instead of “random variable samples.”
    1. I believe the discriminator $\pi(r)$ is used as a surrogate for binary-valued function $\eta$. If so, this could be stated when it is introduced. Accordingly, the “optimal discriminator $\pi^*$” is not optimal for (9) but for its counterpart where 0-1 classification error is replaced by cross-entropy.
    1. Page 8, first displayed equation: I think a ‘+’ is missing as in $\rho 1_p 1’_p + (1-\rho) I_p$. Second displayed equation: The quantity in the exponent is still a vector. Is something missing?
    1. Figure 4: The values reported look too small to be standard deviations. Are they standard errors in the mean?
    1. Appendix A.1, second line: “agree with” —> “make”
    1. Appendix A.1, Definition 3: Should be a comma between $\hat{\tau}_f(x)$ and $\tau(x)$ rather than a minus sign.
    1. Appendix A.2, proof of Lemma 3: In going from (19) to (20), do we also get a Jacobian factor from the change of integration variable from $x$ to $r$, and does that cancel the Jacobian term from the change of variable in the pdfs?

---

> ### Author Response · Authors · 2023-07-22
> **Response to Reviewer LMUR (1/4)**
>
> Dear reviewer LMUR:
>
> We genuinely appreciate your time and effort in providing valuable feedback, and your thoughtful reviews are so helpful and constructive in helping improve our paper! We have carefully considered your suggestions and made corresponding changes. The revised manuscript has been uploaded and the changes are colored in ```RED```. Below, we address the questions and concerns you raised one by one:
>
> * Correctness:
>
> 1. Thank you for pointing out the inconsistency regarding the assumption. We have carefully considered the mathematical details and made a modification to Assumption 1 (before Lemma 1 in Appendix) as follows:
>
> Assumption 1: There exists a constant $K>0$ such that $\underset{y_{2}\in\mathcal{Y},\mathbf{x}\in\mathcal{X}}{\sup}\int_{\mathcal{Y}}L(y_{1},y_{2})p(y_{1}|\mathbf{x})d y_{1}\leq K$.
>
> We believe that this revised assumption aligns appropriately with all the theoretical results presented in the paper. We are grateful for your constructive suggestion!
>
> 2\&3. Thanks for pointing out these typos! We have addressed them based on your detailed feedback.
>
> * Presentation of Theoretical Results and Relationship to Prior Work:
>
> 1. Thank you for pointing out the similarity between Lemma 1, Theorem 1 in the main text, and Lemma 5 in the appendix with Shalit et al. (2017). For the sake of completeness in our theory, we find it necessary to restate these theorems. To address this, we have explicitly mentioned in blue that these results follow Shalit et al. (2017) in the paragraphs immediately following each of these theorems. We really appreciate your insight regarding the connections with related works!
>
> 2. Your suggestion to incorporate the complete theorem from the appendix into the main text is constructive to our paper. This modification allows us to place greater emphasis on our contribution and enhances the overall consistency of the paper. Accordingly, we have made the necessary adjustments to the content (mainly theorem 1 and theorem 2), better highlighting the originality and significance of our work.
>
> 3. Thank you for your comment. Instead of saying “our theoretical derivations are very different from …”, we have rephrased the statement as follows: “our theoretical derivations are not trivial extensions from other related works”. We believe this revision accurately present our theoretical contributions without overclaiming our novelty.
>
> 4. We are grateful for your suggestion of emphasizing the specific contribution of propensity confusion in Introduction. To highlight this aspect, we have emphasized this point in the third paragraph of Introduction. Furthermore, we have provided more detailed explanations in Section 3, particularly in the first paragraph of Section 3.2, to underscore the significance of this notable contribution.

---

> ### Author Response · Authors · 2023-07-22
> **Response to Reviewer LMUR (2/4)**
>
> * Experiments
>
> 1.	We are thankful for your feedback regarding the hyperparameter settings. While it is true that different models can be trained with different hyperparameters. We would like to emphasize that a more comprehensive range of hyperparameters actually exists in addition to $\alpha_1$ and $\alpha_2$. For example, hyperparameters also include the number/units of representation layers, the number/units of outcome layers, learning rate, batch size, and more. To ensure a thorough exploration of the hyperparameter space and find the optimal combination, it is inevitable to search across all possible sets of hyperparameters, i.e., {$\alpha_1$} $\times$ {$\alpha_2$} $\times$ {batch} $\times$ {learning rate} $\times$ {the number (or units) of representation layers} $\times$ {the number (or units) of outcome layers}, etc. Even though we don’t consider the numerous combinations for the number and units of hidden layers and only consider the six hyperparameters, each with two choices (e.g., {learning rate}={0.01, 0.001}), there is a total of 64 combinations for a single model. Given the complexity of hyperparameter searching, we believe that it is fair to maintain fixed hyperparameters across model structures. This allows us to conduct fair ablation studies of the PPBR and PDIG components. Besides, we also would like to highlight that (Curth, A., & van der Schaar, M. (2021)) also mentioned that “to make a fair comparison, we fix hyperparameters across all models” in Section 5 of their paper. It is worth noting that assessing various models is an important area of research, and no universal solution exists to facilitate a perfect comparison among them. This matter is related to key subjects in machine learning, such as model evaluation, model bias, and model fairness. As assessing the performance of different models remains a challenge that requires continuous investigation, we think fixing the parameters in our current version is a relatively fair way to make comparisons and conduct ablation studies.
>
> 2.	Thank you for your suggestion regarding the significance analysis. To assess the significance of the improvements observed in the part of the source of gain, we conducted an additional analysis by recording the values of $\sqrt{\epsilon_{PEHE}}$ and $\epsilon_{ATE}$ for 30 runs of each of the 5 models (GNet, INet, DGNet, DINet, DIGNet). Subsequently, we performed a t-test for GNet-DGNet, INet-DINet, DGNet-DIGNet, and DINet-DIGNet, to investigate the statistical significance of the differences:
>
> |              | Training set             |        |                  |        | Test set                 |        |                  |         |
> |--------------|--------------------------|--------|------------------|--------|--------------------------|--------|------------------|---------|
> |              | $\sqrt{\epsilon_{PEHE}}$ |        | $\epsilon_{ATE}$ |        | $\sqrt{\epsilon_{PEHE}}$ |        | $\epsilon_{ATE}$ |         |
> |              | t                        | p_t    | t                | p_t    | t                        | p_t    | t                | p_t     |
> | GNet-DGNet   | 2.7435                   | 0.0081 | 2.9844           | 0.0042 | 2.7073                   | 0.0089 | 2.9269           | 0.0049  |
> | INet-DINet   | 4.0812                   | 0.0001 | 3.5222           | 0.0008 | 3.5665                   | 0.0007 | 3.0824           | 0.0031  |
> | DGNet-DIGNet | 2.0240                   | 0.0476 | 1.8888           | 0.0639 | 2.0650                   | 0.0434 | 2.0935           | 0.0407  |
> | DINet-DIGNet | 0.4513                   | 0.6535 | 1.3525           | 0.1815 | 0.6079                   | 0.5456 | 1.5473           | 0.1272  |
> ||
>
>
> The t-test results indicate that there is a significant difference in general between GNet-DGNet, INet-DINet, and DGNet-DIGNet. However, it is hard to conclude that the difference in DINet-DIGNet is statistically significant. Therefore, to avoid overclaiming any empirical improvements, we have removed the word "overwhelmingly" from our claims.
>
> Thank you for your insightful comments!
>
> * REF:
>
> [1] Curth, A., & van der Schaar, M. (2021). On inductive biases for heterogeneous treatment effect estimation. Advances in Neural Information Processing Systems, 34, 15883-15894.

---

> ### Author Response · Authors · 2023-07-22
> **Response to Reviewer LMUR (3/4)**
>
> * Clarity
>
> 1.	Thank you for your suggestion! We have added Figure 1 in Introduction to provide a visual explanation of how our model, DIGNet, extends the classic model by incorporating the proposed components PPBR and PDIG. More details of the figure are also provided in the caption of Figure 1 for better understanding.
>
> 2.	We apologize for any unclear points. In our context, the term "patterns" refers to characteristics or information utilized in representation learning. For further clarity, we have included more explanations in the caption of Figure 1.
>
> 3. Thanks for your suggestion. In our response to your comment 4 regarding "Presentation of Theoretical Results and Relationship to Prior Work," we have now clarified the concept of propensity confusion in the third paragraph of the Introduction. Additionally, the original content of Section 4.1 stated the connection between propensity confusion and minimizing $\mathcal{H}$-divergence in the paragraph discussing the "objective of INet".
>
> 4.
>
> (1) The concept of propensity confusion means "it is hard to distinguish whether each unit in the representation space is treated or controlled". Following your suggestions, we have removed the word "concept" and clearly stated the definition of propensity confusion in the third paragraph of the Introduction to avoid any misinterpretations.
>
> (2) The term "rationality" refers to the theoretical guarantee of propensity confusion. To achieve representation balancing, the intuitive approach is to attain propensity confusion, meaning that each unit in the representation space is hard to predict as treated or controlled. This can be implemented with a min-max objective, as explained in Section 4.1, in the discussion of "objective of INet." The theoretical guarantee of this empirical implementation is presented in Section 3.2. To prevent any misleading interpretations, we have now explained more about propensity confusion in the third paragraph and the first paragraph of Section 3.2.
>
> (3) Thank you for pointing this out! We have abbreviated Individual Treatment Effect as ITE in the third paragraph of Introduction, where it appears for the first time.
>
> 5. Yes, you are right. Models such as GNet and INet indeed lack decomposition modes. We have revised the phrase to "If models such as GNet and INet do not have decomposition modes" to avoid any misunderstanding. Thank you for your comment!
>
> 6. INet is meaningful and interpretable due to its foundation in propensity confusion. According to the previous responses and our modifications on propensity confusion and its connection to minimization over $\mathcal{H}$-divergence in Introduction and Section 3.2, when there is propensity confusion, i.e., when it is difficult to distinguish whether each unit in the representation space is treated or controlled, the representations are believed to be adequately balanced. For INet, we achieve propensity confusion by minimizing the $\mathcal{H}$-divergence-based ITE error bound (please refer to the implementation details in Section 4.1, in the paragraph discussing the "objective of INet").
>
> 7. To ensure a fair comparison with the baseline models, we report the results of the baseline models on the benchmark dataset as originally reported in their respective papers. However, reproducing the reported values using the code provided by the papers is not always feasible. If relevant results are reported in their original paper, we directly report them. If relevant results are not available in the original paper, and we cannot reproduce them with the provided code as good as the reported values, we use "-" to indicate the results in Table 3. For example, if a paper reports $\epsilon_{ATE}$ but not $\sqrt{\epsilon_{PEHE}}$, and we cannot reproduce $\epsilon_{ATE}$ results as good as their reported ones, we use "-" for $\sqrt{\epsilon_{PEHE}}$.
>
> 8. Thank you for your suggestion. We have updated Lemma 2 in the Appendix as: "Let $\mathcal{G}$ in Definition 9 be the family of binary functions. Then we obtain $\sup_{\eta\in\mathcal{H}}\left|\int_\mathcal{S}\eta(s)(p_1(s)-p_2(s))ds\right|=\frac{1}{2}d_\mathcal{H}(p_1,p_2)$."

---

> ### Author Response · Authors · 2023-07-22
> **Response to Reviewer LMUR (4/4)**
>
> * Minor Comments and Questions
>
> 1. Thank you for your suggestion! We have modified "random variable samples" to "random variables".
>
> 2. Yes, your understanding is correct. We have added an explanation before Eqn. 12, stating that $\pi$ can be regarded as a surrogate for the binary-valued function $\eta$.
>
> 3. We are grateful for your careful correction of our typo! We missed the symbol "+". The correct expression is: $\mathbf{X}_{i}  \sim \mathcal{N} (\mathbf{0}, \sigma^{2}\cdot [ \rho \mathbf{1}_p  \mathbf{1}_p^{'} + (1-\rho) \mathbf{I}_p ])$. We apologize for the oversight.
>
> 4. Yes, you are right. As stated in the captions of the Tables, the reported values are in the format "mean $\pm$ standard error," which is consistent with (Shalit et al., 2017). The standard error is defined as the standard deviation divided by the square root of $N$, where $N$ represents the sample size.
>
> 5\&6. Thank you for your feedback. We have made the necessary modifications according to your suggestions.
>
> 7. Yes, you are right. To clarify the equality from (19) to (20), we provide detailed derivations as follows:
>
> Let $\mathbf{x}=\Psi(\mathbf{r})$. By the *change of variables for pdfs*, we have $p^{T=0}_ {\Phi} (\mathbf{r})=p^{T=0}(\mathbf{x})|J_\Psi(\mathbf{r})|$, $p^{T=1}_ {\Phi}(\mathbf{r})=p^{T=1}(\mathbf{x})|J_\Psi(\mathbf{r})|$, where $J_\Psi(\mathbf{r})$ is the determinant of the Jacobian matrix, and $|\cdot|$ is the absolute operator. Now, by the *method of substitution* (i.e., $\mathbf{x}=\Psi(\mathbf{r})$), we have
>
> $\int_{\mathcal{X}} \ell_{h,\Phi} (\mathbf{x},0) (p^{T=1}(\mathbf{x})-p^{T=0}(\mathbf{x})) d\mathbf{x}=\int_{\mathcal{R}} \ell_{h,\Phi}(\Psi(\mathbf{r}),0) [\frac{p^{T=1}_{\Phi}(\mathbf{r})}{|J _{\Psi(\mathbf{r})}|} - \frac{p^{T=0} _{\Phi}(\mathbf{r})}{ |J _{\Psi(\mathbf{r})}| }] |J _{\Psi(\mathbf{r})}|d\mathbf{r}.$
>
> As we can see, the term $|J_\Psi(\mathbf{r})|$ cancels out, leading to:
>
> $\int_{\mathcal{X}} \ell_{h,\Phi}(\mathbf{x},0) (p^{T=1}(\mathbf{x}) - p^{T=0}(\mathbf{x})) d\mathbf{x} =\int_{\mathcal{R}} \ell_{h,\Phi}(\Psi(\mathbf{r}),0) (p^{T=1} _{\Phi}(\mathbf{r}) - p^{T=0} _{\Phi}(\mathbf{r})) d\mathbf{r}$.
>
> Similarly, we have
>
> $\int_{\mathcal{X}} \ell_{h,\Phi}(\mathbf{x},1) (p^{T=0}(\mathbf{x}) - p^{T=1}(\mathbf{x})) d\mathbf{x} =\int_{\mathcal{R}} \ell_{h,\Phi}(\Psi(\mathbf{r}),1) (p^{T=0} _{\Phi}(\mathbf{r}) - p^{T=1} _{\Phi}(\mathbf{r})) d\mathbf{r}$.
>
> Finally, we would like to express our gratitude to the reviewer again for your detailed, careful, thoughtful, and constructive feedback! We hope that our responses and the revised paper have adequately dispelled your doubts. We remain receptive to further discussions and appreciate any additional comments or suggestions you may have.

---

### Author Response · Authors · 2023-08-29
**We are looking forward to your feedback!**

Dear Reviewers,

We want to express our sincere appreciation for the time and effort you and the Action Editor have dedicated to reviewing our paper, and we truly appreciate your help with the refinement of our research.

As you may know, our paper has been under review for more than five months, and we understand that you may have been occupied with your demanding schedules. With the completion of our responses, we kindly request your assistance in finalizing the review process at your earliest convenience.

We greatly value your insights and expertise, and we are very open to any further comments, questions, or suggestions you may have. We are looking forward to your replies!


Best regards,

Authors of Paper949

---

### Decision · Action_Editors · 2023-09-04

**Recommendation:** Reject

**Comment:**

In their final assessments, two reviewers provided recommendations of "leaning reject" while one provided a recommendation of "leaning accept".  The general sentiment was that the submission shows good promise and should be publishable in the future, but further work is still needed to get it to this point, while the amount of significant changes made since the original submission is reaching the point where a re-review would be required regardless.  I generally concur with this majority viewpoint and though I believe this is quite a borderline decision, my recommendation is for the paper to be rejected with a strong encouragement to submit a major revision.

For this major revision, I suggest further addressing the concerns raised in the "Claims and Evidence" section.  One possible exception to this is the introduction of theory for when the methods are combined: though this would be an excellent addition, I appreciate that it might not be viable to add in practice and I do not think the lack of this undermines the publication of the paper.  I would also encourage the authors to undertake further work on the manuscript as per the many other more specific reviewer comments.

One extra point of note here is that I believe there is still a lot more that could be done to improve the clarity of the work, on top of the already significant (and helpful) updates that have been made in the discussion period.  I generally found the submission quite difficult to follow and often struggled to separate what aspects were new contributions compared with previous work and established concepts.  I also often found it difficult to match up the theory in the Appendix with the main paper (e.g. the Theorem's don't quite match those in the main paper, the theory in the Appendix is not directly presented in the form of proofs of the Theorems in the main paper, and its difficult to see where to look when going from the main paper to find the appropriate part of the Appendix).  As well as being a problem in its own right, this also made it difficult to assess where the updated assumption was sufficient to correct the proof of the quoted results.

**Audience:**

Though there is some disagreement among the reviewers about the precise level of novelty and significance of the work, the consensus is that the work will be of interest to some in the community; all reviewers agreed that the submission meets this requirement for TMLR.

**Claims And Evidence:**

In their final recommendations, one reviewer felt the paper met the required threshold with claims and evidence, while the other two did not.  The key concerns of the latter were:
- Whether correctness issues with the theoretical results have been fully resolved (especially regarding Assumption 1 in the Appendix, with, for example, Theorem 2 still stating the old assumption rather than the new one)
- Failures in properly distinguishing from previous work and making clear what aspects of the work are new are what are not.  For example, it is still not entirely clear in the paper itself how much of Theorem 1 should already be credited to Shalit et al and what (non-trivial) aspects have been added.  Personally, I also found it difficult more generally to distinguish between what was prior work and was new contribution throughout the paper.
- Whether key experimental comparisons were made fairly when the hyperparameter tuning may well be biasing towards the introduced method.
- Overclaiming about some of the results when they do not appear to pass a statistical significance test (corrections and new tests were suggested for this in this discussion, but do not seem to have been properly transferred to the paper).
- The lack of any theoretical analysis when the proposed methods are combined.
- General concerns about the clarity of the paper.  My own assessment of this was also quite negative, perhaps more so than the reviewers.

**Resubmission Of Major Revision:**

The authors may consider submitting a major revision at a later time.